# ExoPredicator: Learning Abstract Models of Dynamic Worlds for Robot Planning

**Yichao Liang**[1,2], **Dat Nguyen**[2], **Cambridge Yang**[2], **Tianyang Li**[2], **Joshua B. Tenenbaum**[5],

**Carl Edward Rasmussen**[1], **Adrian Weller**[1,6], **Zenna Tavares**[2], **Tom Silver**[4,†], **Kevin Ellis**[3,†]

[1]University of Cambridge, [2]Basis, [3]Cornell University, [4]Princeton University,
[5]Massachusetts Institute of Technology, [6]The Alan Turing Institute, [†]co-advising

## Abstract

Long-horizon embodied planning is challenging because the world does not only change through an agent's actions: exogenous processes (e.g., water heating, dominoes cascading) unfold concurrently with the agent's actions. We propose a framework for abstract world models that jointly learns (i) symbolic state representations and (ii) causal processes for both endogenous actions and exogenous mechanisms. Each causal process models the time course of a stochastic cause-effect relation. We learn these world models from limited data via variational Bayesian inference combined with LLM proposals. Across five simulated tabletop robotics environments, the learned models enable fast planning that generalizes to held-out tasks with more objects and more complex goals, outperforming a range of baselines.

## 1 Introduction

For an agent to think about the future consequences of its actions, does it need to simulate the world pixel-by-pixel, frame-by-frame, or can it reason more abstractly? Consider planning a flight to another country: we can reason about buying tickets, changing airplanes, and crossing borders without committing to the color of the airplane or the milliseconds before takeoff. Absent abstraction, planning over long time horizons would be intractable, because every minute detail of the world would need to be simulated. This intuition is captured by *abstract world models*, (Konidaris, 2019; Wong et al., 2025) which retain information essential for decision-making, while hiding irrelevant details.

Recent work on learning abstract world models (e.g., (Liang et al., 2024; Athalye et al., 2024)) assumes that the world changes only by direct, instantaneous actions. But in the real world, our actions are not instantaneous, and are only half the story: the external world has its own causal mechanisms, which unfold continuously in time concurrent with our own actions. For instance, consider boiling water (Figure 1). After switching on a kettle, the water's temperature continuously rises independently of the agent's subsequent actions until it finally boils. *A good abstract world model must therefore abstract not just the states, but also the temporal granularity:* decision-makers should know that switching on a kettle triggers another causal mechanism, which eventually results in boiling, without reasoning about the exact timecourse of the water's temperature (i.e., a robot could chop vegetables while waiting for the water to boil). Such abstraction is conceptually separate from options/skills/high-level actions, which abstract the timecourse of one's own actions, but do not abstract the timecourse of external causal processes in the outside world.

The standard planning representation used for decades, PDDL, also fails to capture this: classical PDDL treats actions as instantaneous; while PDDL 2.1(Fox & Long, 2003) adds durative actions, it still does not explicitly model autonomous exogenous processes without extensions such as PDDL+(Fox & Long, 2002). Learning a symbolic PDDL planning model (Silver et al., 2023; Liang et al., 2024) combinatorially explodes with respect to temporal granularity. Vision-language models (VLMs) and vision-language-action models (Team et al., 2023; Black et al., 2025) could in princi-

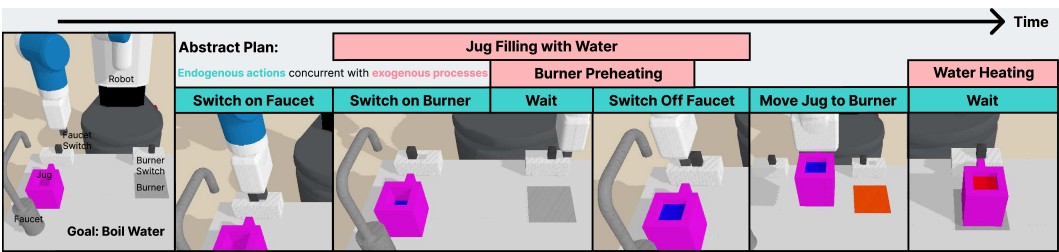

Figure 1: *Dynamic* environments include both *endogenous* processes (actions under the agent's direct control, such as **Switch On Faucet**) and *exogenous* processes (e.g., **Jug Filling with Water**) that evolve on their own. Planning requires reasoning about both kinds of processes.

ple reason about external causal mechanisms, but generalize poorly to novel situations, particularly when reasoning about temporal physical constraints.

To address these challenges, we introduce a framework for learning world models that abstract both the state space, and the timecourse of causal processes. We contribute the following: (1) A symbolic yet learnable representation of abstract world models for environments with temporal dynamics and external causal processes. (2) A state abstraction learner that leverages the commonsense knowledge of foundation models. (3) An efficient Bayesian inference method for learning the parameters and structures of these causal models. (4) A fast planner for reasoning with the proposed representation.

## 2 BACKGROUND AND PROBLEM FORMULATION

We consider learning abstract world models for robot planning in environments whose causal mechanisms include both the agent's own action space, and external mechanisms not directly under the agent's control. The actual environment operates frame-by-frame (high temporal granularity), and exposes a state space with object tracking features and pixel-level visual appearance (high-resolution perception). We assume built-in motor skills, such as `Pick`/`Place`, a common assumption (Kumar et al., 2024; Silver et al., 2021). The goal is to learn a world model abstractly describing the timecourse of causal processes, and to generalize to held-out decision-making tasks.

**Environments.** An environment $\mathcal{E}$ is a tuple $\langle \mathcal{X}, \mathcal{U}, \mathcal{C}, f, \Lambda \rangle$ where $\mathcal{X}$ is a state space, $\mathcal{U} \subseteq \mathbb{R}^m$ is a low-level action space (e.g. motor torques), $\mathcal{C}$ is a set of controllers for skills (e.g. `Pick`/`Place`), $f : \mathcal{X} \times \mathcal{U} \to \mathcal{X}$ is a transition function, and $\Lambda$ is a set of *object types* (object classifier outputs).

**Tasks.** Within an environment, a task $T$ is a tuple $\langle \mathcal{O}, x_0, g \rangle$ of objects $\mathcal{O}$, initial state $x_0$, and goal $g$. The allowed states depend on the objects $\mathcal{O}$, so we write the state space as $\mathcal{X}_{\mathcal{O}}$ (or sometimes just $\mathcal{X}$ when the objects are clear from context). Each state $x \in \mathcal{X}_{\mathcal{O}}$ includes associated object features, such as 3D object position. The environment is shared across tasks.

## 3 ABSTRACTING STATES, TIME, AND CAUSAL PROCESSES

Environments present a high-dimensional observation space that evolves frame-by-frame. Abstract world models hide this complexity behind a *state abstraction*, which distills a small set of features from observations, together with *causal processes*, which describe temporal dynamics (Figure 2).

**State Abstraction with Predicates.** A predicate $\psi$ – after being parameterized by specific objects – is a Boolean feature of states. A predicate is represented as a Python function that queries a VLM to inspect the robot's visual input. We treat this as function $\psi : \mathcal{O}^m \to (\mathcal{X} \to \mathbb{B})$ that given $m$ objects predicts whether a predicate holds in a state. A set of predicates $\Psi$ induces an *abstract state* $s$ corresponding to all the predicate/object combinations (*ground atoms*) that hold in that state:

$$s := \text{ABSTRACT}_{\Psi}(x) = \{(\psi, o_1, ..., o_m) \; : \; \psi(o_1, ..., o_m) \text{ holds in state } x, \text{ for } \psi \in \Psi \text{ and } o_j \in \mathcal{O}\}$$

Thus an abstract state is the collection of symbolic facts the agent believes about the world. This style of state abstraction is standard in robot task planning (Garrett et al., 2021). We write $\mathcal{S}$ for the set of possible abstract states.

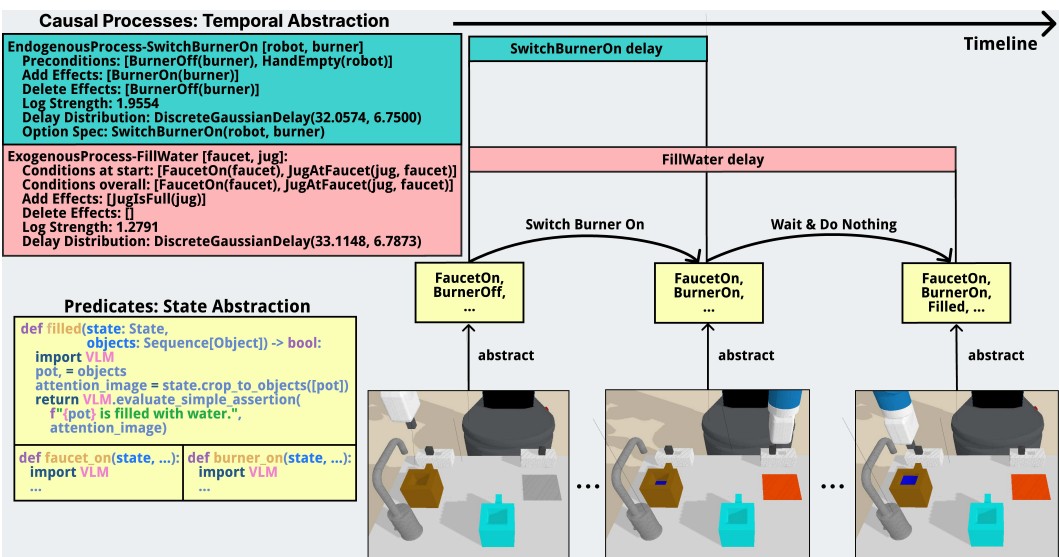

Figure 2: Raw input maps to a state abstraction via *predicates*: short Python programs detecting binary features. Learning the state abstraction means synthesizing these programs. Temporal dynamics of abstract states are governed by *causal processes*: either *endogenous processes* (actions), or *exogenous processes* in the outside world. Causes realize their effects only after a delay, and can be interleaved. Learning causal processes allows planning by breaking frame-by-frame dynamics into discrete jumps between abstract states. (Illustration simplified; see text.)

**Causal Processes: Informal intuition.** A causal process coarsely models a cause-effect relation. For example, opening a faucet above a bucket triggers a chain of cause-effect relations: a stopper opens, hidden pipes fill with water, water rises in the bucket, and eventually the bucket is filled. For decision making, many such details are irrelevant, or omitted from the state abstraction. A causal process therefore abstracts away such details by saying that certain conditions (the "cause") later lead to other conditions (the "effect"). We further distinguish two kinds of causal processes. **Endogenous processes** correspond to the agent's high-level actions or skills. They represent operations that are under the agent's direct control, such as switching a faucet on. **Exogenous processes** describe the background dynamics of the environment. They represent processes that unfold autonomously once certain conditions hold, independent of whether the conditions were created by the agent or by some external mechanism, such as a kettle filling with water after being placed under a running tap. This separation allows the agent to reason about the consequences of its actions while also anticipating changes initiated by the environment itself. Figure 2 (top right) shows concrete examples of such causal processes (e.g., switching a burner on, filling a jug), with their conditions, delayed effects, and whether they are endogenous or exogenous.[1]

**Causal Processes: Formalization.** Abstract world models are equipped with a set of causal processes $\mathcal{L}$. Each causal process $L \in \mathcal{L}$ is defined by a **schema** tuple, $\langle \text{PAR}, C, O, E, W, p^{\text{delay}} \rangle$ (see fig. 2, top right, for concrete schema examples), where:

- PAR (parameters) is a list of typed variables present in the condition or effect of the process.

- $C$ is the *condition at start*, a set of atoms that must be true for the process to be activated.

- $O$ is the *condition overall*, a set of atoms that must remain true throughout the process's duration.

- $E$ is the *effect*, an add and a delete set of atoms describing the state change upon completion.

---

[1]Our terminology differs from standard definitions in causality (Pearl, 2009) and recent RL literature (e.g., Efroni et al. (2021)), where "exogenous" typically refers to variables that are determined outside the system or are irrelevant to the task. In contrast, our exogenous processes are task-relevant and can be triggered by the agent (and are thus causally downstream), but differ in that they evolve autonomously without requiring continuous agent actuation. This aligns with the definition of "processes" in temporal planning frameworks like PDDL+ (Fox & Long, 2002)

- $W \in \mathbb{R}$ (log strength) quantifies how likely the effect will happen when the conditions are satisfied.
- $p^{\text{delay}}$ is a probability distribution over the delay between the process's activation and effect (Appendix C.3).

Endogenous processes further include a skill $c \in \mathcal{C}$. A schema is instantiated into a **ground causal process** $\underline{L} = \langle \underline{\text{PAR}}, \underline{C}, \underline{O}, \underline{E}, W, p^{\text{delay}} \rangle$ (and optionally $\underline{c}$) by substituting its parameters with specific objects. For example, the endogenous process `SwitchBurnerOn` has parameters `?robot` and `?burner`. Its condition $C$ requires that the burner is off and the robot's hand is free. Once executed, its effect $E$ (the burner being on) occurs after a delay sampled from $p^{\text{delay}}$.

**Interdependence of state abstraction and causal processes.** The right abstractions depend on downstream tasks, as our goal is generalization to unseen planning problems. Tasks demanding a more detailed state abstraction will generally require modeling more temporal dynamics. This couples the causal processes to the state abstraction.

**Probabilistic Semantics.** The causal processes define how abstract states evolve over time. Mathematically, $\mathcal{L}$ defines a probabilistic generative model over sequences of abstract states $\{s_t\}$ at fine-grained timesteps $t \in \mathbb{N}$. At each step, active causal processes contribute pressure for their effect atoms to change, while a separate frame term prefers all atoms to remain as they are. To write this compactly, we use indicator functions for the *conditions at start*: $C_{\underline{L}}(s_{1:t})$ is 1 iff all atoms in the condition-at-start set $C$ of ground process $\underline{L}$ are satisfied at state $s_t$ but not satisfied at state $s_{t-1}$, and 0 otherwise; similarly, $O_L(s_{t'+1:t-1})$ is 1 iff all atoms in the overall-condition set $O$ hold at every step between $t' + 1$ and $t - 1$. We probabilistically model the effect of ground process $\underline{L}$ as a potential $E_{\underline{L}}^j$ upon each feature $j$ of the abstract state:

$$\log E_{\underline{L}}^j(s^j) = \begin{cases} W_{\underline{L}} \times s^j & \text{if } j \in E_{\underline{L}}.\text{Add} \\ W_{\underline{L}} \times (1 - s^j) & \text{if } j \in E_{\underline{L}}.\text{Del} \\ 0 & \text{otherwise.} \end{cases}$$

We similarly define a *frame axiom potential* $\log F(s_t^j | s_{t-1}^j) = W_F \times 1\left[s_t^j = s_{t-1}^j\right]$, which encourages states to stay constant over time; $W_F$ is a learnable parameter. Because causal processes have stochastic delays, we associate each ground causal process $\underline{L}$ with random variables $\{\Delta_t^L\}$ for the delay should the process trigger at time $t$, i.e. $\Delta_t^L \sim p_{\underline{L}}^{\text{delay}}$. With these definitions in hand, the joint distribution over delays and abstract states is

$$p(s_{1:t}, \{\Delta_{1:t}^L\} \mid s_0) = \prod_t p\left(s_t \mid s_{<t}, \{\Delta_{<t}^L\}\right) \prod_{\underline{L}} p(\Delta_t^L | s_{\leq t}) \qquad \text{(autoregressive)}$$

$$p\left(s_t \mid s_{<t}, \{\Delta_{<t}^L\}\right) = \prod_j p\left(s_t^j \mid s_{<t}, \{\Delta_{<t}^L\}\right) \qquad \text{(next-state factorizes over features)}$$

$$p\left(s_t^j \mid s_{<t}, \{\Delta_{<t}^L\}\right) \propto F(s_t^j | s_{t-1}^j) \prod_{\substack{\underline{L} \in \mathcal{L} \\ t' < t}} E_{\underline{L}}^j(s_t^j)^{C_{\underline{L}}(s_{1:t'}) O_{\underline{L}}(s_{t'+1:t-1}) 1[t = t' + \Delta_{t'}^L]} \qquad \text{(cause-effect)}$$

$$p(\Delta_t^L | s_{\leq t}) = p_{\underline{L}}^{\text{delay}}(\Delta_t^L)^{C_{\underline{L}}(s_{1:t})} \qquad \text{(delay distribution for when condition at start holds)}$$

But this formalization simulates every fine-grained timestep: reasoners should abstract away temporal details like the milliseconds before a domino falls. We describe next how to do that.

## 4 PLANNING WITH CAUSAL PROCESSES

We model the world as changing abruptly in discrete "jumps" between abstract states. Planning can clump together stretches of time where the abstract state remains constant. We define a **big-step transition function**, $\mathcal{T}_{\text{big}}$, which runs the world model until the abstract state changes, and optionally takes as input an action to initiate. The agent doesn't always need to manipulate an object directly; it can also choose to wait for an exogenous process to unfold on its own (such as waiting for water to

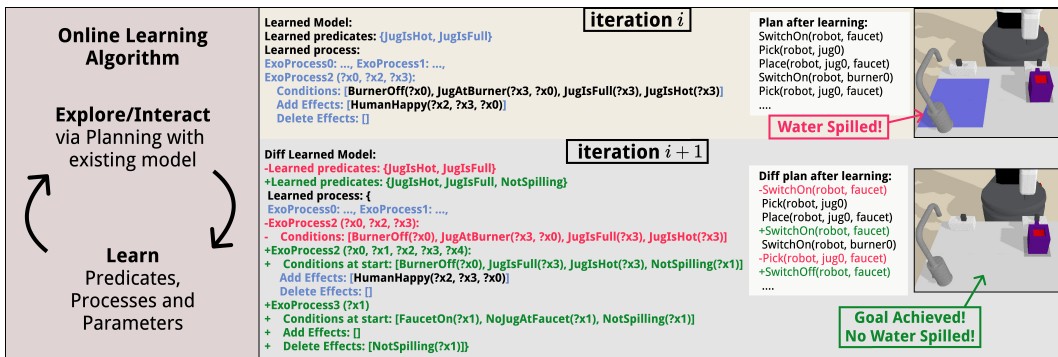

Figure 3: The online learning loop, where the agent repeatedly uses its current model to plan and interact with the world, then refines that model by learning new predicates and causal processes from the experience. The figure shows an example where the agent's initial model in iteration $i$ leads to a failed plan (Water Spilled!). After observing this failure and updating its knowledge ("Diff Learned Model"), the agent creates a successful plan in iteration $i + 1$ ("Diff plan after learning").

boil). The agent achieves this by initiating a special `NoOp` (no operation) action, which terminates as soon as the abstract state changes. Formalizing $\mathcal{T}_{\text{big}}$ is relatively technical; see Appendix A.1.

This big-step function allows a planner to perform forward search in the space of high-level actions, simulating the concurrent and delayed effects of both its own actions and the environment's exogenous dynamics. Given causal processes $\mathcal{L}$ and a task, the agent performs an A* search over sequences of ground endogenous processes. The search uses $\mathcal{T}_{\text{big}}$ to determine successor states and we design a version of the fast-forward heuristic (Hoffmann, 2001) to guide the search (Appendix A.2).

## 5 PROCESS LEARNING AND PREDICATE INVENTION

Our goal is to learn how the outside world works: we assume an unfamiliar environment, but not an unfamiliar body. We therefore equip the agent with some basic predicates (such as whether it is holding an object) and endogenous causal processes defining its own action space, such as `Pick`/`Place` (Appendix B), and learn the remaining causal processes and state abstractions.

We initialize with 1-2 demonstration trajectories, and then perform online learning (Figure 3). Online learning involves planning to solve training tasks to collect further trajectories (when planning fails, random actions are taken). At each stage of online learning, we have a dataset of state-action trajectories $\mathcal{D}_{\text{low}} = \{(x_0, \underline{c}_0, \ldots, \underline{c}_{T-1}, x_T)\}$. Given predicates $\Psi$, this generates a dataset of abstract state-action trajectories $\mathcal{D}_{\text{abs}} = \{(s_0, \underline{c}_0, \ldots, \underline{c}_{T-1}, s_T)\}$. Given this dataset, we learn predicates (Section 5.3), exogenous processes (section 5.2), and continuous parameters (Section 5.1)—described in a reverse order, because each component is used in the previous component during learning.

### 5.1 PARAMETER LEARNING VIA VARIATIONAL INFERENCE

Each causal process has continuous parameters for the delay distribution $p_L^{\text{delay}}$ and the probabilistic weight $W_L$, and we have a global parameter $W_F$ for the frame axiom. Fixing all discrete structure—including the predicates—we seek parameters maximizing the marginal likelihood $p(\mathcal{D}_{\text{abs}}) = p(s_{1:t}|s_0) = \sum_{\{\Delta_{1:t}^L\}} p(s_{1:t}, \{\Delta_{1:t}^L\}|s_0)$. This is intractable because there are combinatorially many ways of timing the cause-effects.

We therefore approximate the marginal likelihood by introducing variational distributions over the time at which each process realizes its effect: categorical distribution $q_t^L(A_t^L)$ encodes belief about $A_t^L$, the "arrival time" of the effect coming from the process $\underline{L}$ due to its activation at time $t$. This is a change of basis from delays to absolute times, $A_t^L = \Delta_t^L + t$, which leads to a tidier ELBO decomposition. We provide the full expression and derivation of the following variational lower

bound on the marginal likelihood in Appendix A.3, which we optimize using Adam (Kingma, 2014).

$$\log p(s_{1:t}) \geq \underbrace{\mathbb{E}_q\Big[\log p\big(\{A_t^L\}\big)\Big]}_{\text{delay model}} + \underbrace{\mathbb{E}_q\Big[\log p\big(\{s_t\} \mid \{A_t^L\}\big)\Big]}_{\text{abstract dynamics}} + \underbrace{\mathbb{H}_q\Big[\{A_t^L\}\Big]}_{\text{entropy regularizer}}$$

## 5.2 LEARNING EXOGENOUS PROCESSES: BAYESIAN MODEL SELECTION, LLM GUIDANCE

We learn causal processes by assuming a fixed set of predicates $\Psi$ (which fixes the abstract state space $\mathcal{S}_\Psi$): we **segment** the trajectories into shorter clips, where each clip consists of a sequence of constant abstract states followed by a final state in which one or more atoms change value; **cluster** segments according to which features in the abstract state were changed; then **learn** one process per cluster by optimizing Bayesian criteria. To make optimization tractable, we use an LLM to propose different symbolic forms for processes and then score them with our Bayesian objective.

Given a set of trajectories $\mathcal{D}_{\text{abs}}$, we would ideally learn the causal processes $\mathcal{L}^\star$ maximizing

$$\mathcal{L}^\star = \arg\max_{\mathcal{L}} p(\mathcal{L}|\mathcal{D}_{\text{abs}}) = \arg\max_{\mathcal{L}} p(\mathcal{D}_{\text{abs}}|\mathcal{L})p(\mathcal{L}) \qquad \text{(intractable)}$$

where $p(\mathcal{L})$ is a minimum description length prior and $p(\mathcal{D}_{\text{abs}}|\mathcal{L})$ is approximated by Section 5.1. As this optimization is intractable, we learn a separate process $L_\mathcal{C}$ for each cluster $\mathcal{C}$:

$$\mathcal{L}^\star = \bigcup_{\mathcal{C}\in\text{CLUSTER}(\mathcal{D}_{\text{abs}})} \{L_\mathcal{C}\} \quad \text{where } L_\mathcal{C} = \arg\max_L p(\mathcal{C}|L)p(L) \qquad \text{(still intractable)}$$

But computing $\arg\max_L p(\mathcal{C}|L)p(L)$ means optimizing over combinatorially many discrete structures for $L$. To narrow down the discrete search, we prompt a language model with the cluster and ask it to propose a small number of candidate processes:

$$\mathcal{L}^\star = \bigcup_{\mathcal{C}\in\text{CLUSTER}(\mathcal{D}_{\text{abs}})} \{L_\mathcal{C}\} \quad \text{where } L_C = \arg\max_{L\in\text{PROMPT}(\mathcal{C})} p(\mathcal{C}|L)p(L) \qquad \text{(tractable)}$$

Appendix A.4 fully specifies this algorithm, which builds on Chitnis et al. (2022).

## 5.3 LEARNING STATE ABSTRACTIONS: PROGRAM SYNTHESIS AND LOCAL SEARCH

The abstract state space is defined by a collection of short Python programs (predicates) which check for an abstract feature within the raw perceptual input. Learning the state abstraction therefore means synthesizing that set of programs. One strategy for learning the predicates is to *propose* a large set of candidate predicates and then use discrete search to *select* a subset optimizing certain objectives (Silver et al., 2023) In our setting, we propose predicates by prompting an LLM with trajectories, and seek a subset of those predicates, $\Psi^\star$, maximizing:

$$\Psi^\star = \arg\max_{\Psi\in\mathcal{P}(\text{PROMPT}(\mathcal{D}_{\text{low}}))} p(\mathcal{D}_{\text{low}} \mid \mathcal{L}^\star(\Psi))p(\mathcal{L}^\star(\Psi))p(\Psi)$$

where $\mathcal{P}(\cdot)$ is the powerset, $p(\Psi)$ is a prior favoring fewer predicates, and we have made explicit the dependence of $\mathcal{L}^\star$ upon the predicates $\Psi$ (Section 5.2). But the above objective is intractable for two reasons, which we address as follows (Appendix A.5):

- **Expensive outer loop:** The powerset is exponentially large. Rather than exhaustively enumerate every subset of predicates, we do a local hill-climbing search starting from $\Psi = \varnothing$ and greedily adding new predicates from the LLM (Silver et al., 2023).

- **Expensive inner loop:** Scoring a candidate subset of predicates is expensive, requiring variational inference and causal process structure learning (Sections 5.1 and 5.2). We therefore run structure learning and parameter estimation *only once, using all proposed predicates*, and cache the resulting processes and parameters for reuse when scoring different subsets of predicates.

## 6 EXPERIMENTS

We design our experiments to answer the following questions: (**Q1**) How does ExoPredictor perform compared to state-of-the-art methods, including hierarchical reinforcement learning (HRL),

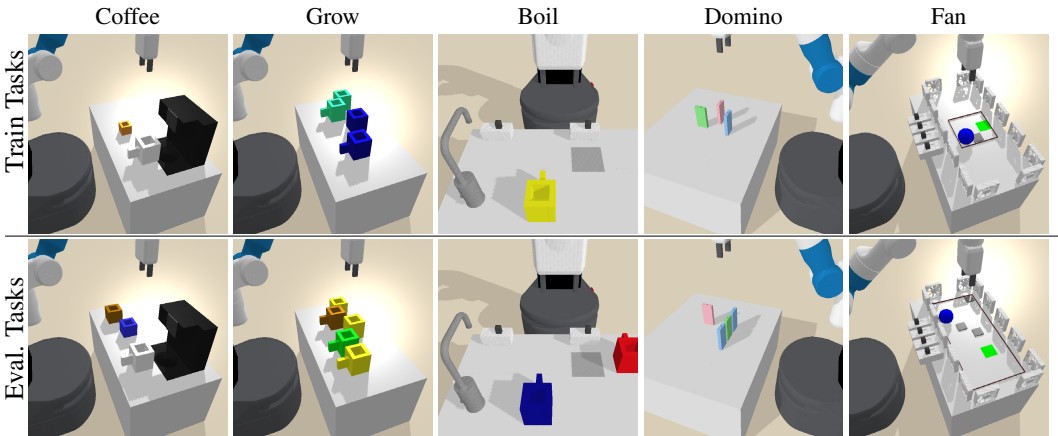

Figure 4: Environments. Top row: train task examples. Bottom row: evaluation task examples.

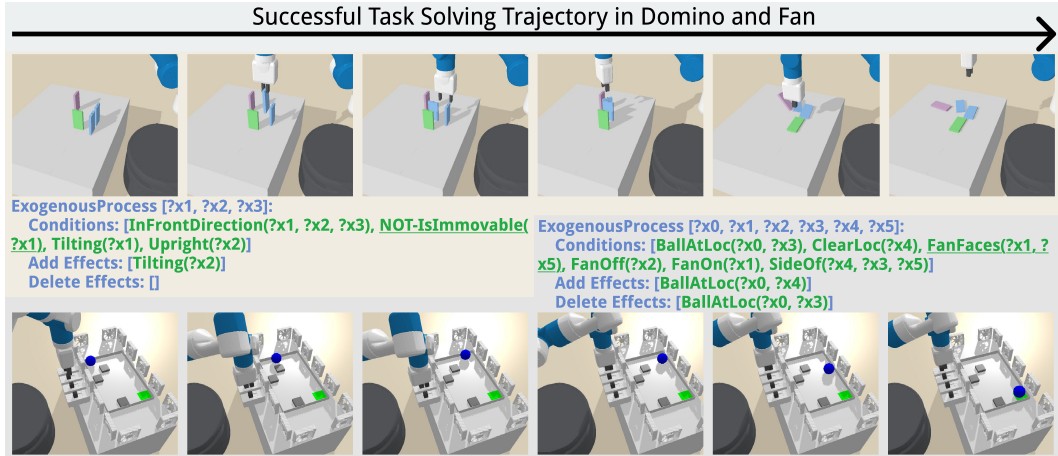

Figure 5: Successful ExoPredicator trajectories in the `Domino` (top) and `Fan` (bottom) environments. The code highlights the key learned exogenous processes, describing how dominoes cascade and how the fan's wind moves the ball. These processes incorporate predicates invented by the agent, like `NOT-IsImmovable` and `FanFaces`, which enable efficient and effective planning.

VLM planning, and operator learning approaches, in terms of overall solve rate and sample efficiency? (**Q2**) How do the learned abstractions perform relative to manually engineered abstractions, and relative to the case where no learning is performed? (**Q3**) How useful are the Bayesian model selection and the LLM-guidance components in model learning?

**Experimental Setup.** We evaluate eight approaches across five simulated robotics environments, illustrated in Figure 4, using the PyBullet physics engine (Coumans & Bai, 2016). All results are averaged across five random seeds. For each seed, we train the agent with one or two training tasks, each of which includes one demonstration. We then evaluate their performance on 50 held-out test tasks, which include more objects and more complex goals. In each online learning iteration, the agent performs 8 rollouts in a training task with each rollout lasting a maximum of 300 timesteps.

**Environments.** We describe the environments and their corresponding predefined closed-loop skills, which are shared across all approaches. All environments have a `NoOp` skill in addition to the ones listed for each environment. See Appendix B for more details.

1. **Coffee.** The agent is asked to fill the cups with coffee. To do so, the agent needs first to get coffee from the coffee machine, then pour it into cups, both of which are exogenous processes. The environment provides 4 skills: `Pick`, `Place`, `Push`, and `Pour`.

2. **Grow.** The agent is tasked with watering plants in pots. A plant will only grow when watered by a jug that has the same color as its pot. The provided skills include `Pick`, `Place` and `Pour`.

3. **Boil.** A cooking domain where the agent is asked to fill jugs with water using the faucet and boil it with the burner, without overspilling any water, which may happen when the faucet is on and no jug is under it or the jug underneath is full, which are all exogenous processes. Four skills are defined in this domain, including `Pick`, `Place`, and `Switch On`/`Off`.

4. **Domino.** A domino puzzle environment with two types of tasks. The agent is tasked to only move the blue dominos and push the green dominoes such that all the purple target dominoes are toppled. Additionally, there are "impossible" tasks, where there are red target dominoes whose mass is too large to be toppled in a cascade. Impossible tasks are "solved" if the agent predicts that the goal is unachievable. Inter-domino dynamics are exogenous processes. The included skills are `Pick`, `Place`, and `Push`.

5. **Fan.** A maze environment where a ball is blown by fans. The agent must control the fans in each cardinal direction to move the ball to the green target location while avoiding obstacles. The provided skills are turning the `Switch On`/`Off`.

**Approaches.**

1. **Manual**. A planning agent with manually engineered predicates and processes for each domain.

2. **Ours**. Our ExoPredicator approach.

3. **MAPLE** (Nasiriany et al., 2022). An HRL baseline that learns to select ground controllers by learning an action-value function, but does not explicitly learn abstract world models and perform lookahead planning. This approach is provided with 1000 training tasks and given a budget of 10000 interaction rollouts per online learning iteration.

4. **ViLA** (Hu et al., 2023). A VLM planning baseline that prompts a VLM (Gemini-2.5-Pro) to plan a sequence of ground skills. We experiment with two variants, which either exclude (zero-shot; zs) or include (few-shot; fs) the demonstrations. Both this and MAPLE are provided with the full set of predicates in **Manual**, which they can use as termination conditions for the `NoOp` action.

5. **VisPred** (VisualPredicator) (Liang et al., 2024). An online STRIPS-style operator learning and planning agent. We provide it with all the necessary predicates used in **Manual**, which sidestep the challenge of predicate learning, to highlight the difference between our *causal processes* representation with traditional STRIPS-style representations.

6. **No Bayes**. An ablation that uses an LLM to learn processes without Bayesian model selection.

7. **No LLM**. An ablation that replaces the LLM condition proposer with a fast-to-compute heuristic. Note that we still use an LLM for predicate proposal.

8. **Manual-d (Manual minus tuned delay parameters)**. A planning agent that uses the **Manual** abstraction, but has the same delay parameter (e.g., 1) across all processes.

9. **No invent**. An ablation that uses the initial abstractions and does not perform any learning.

**Results and Discussion.** Figure 6 shows the evaluation solve rate for all approaches and the learning curve of our online learning agents.

**(Q1).** Our approach consistently outperforms the VLM planning (**ViLa**), HRL (**MAPLE**) and STRIPS-style operator learning and planning (**VisPred**) approaches, achieving a near-perfect score across all domains. For each environment, ExoPredicator learns 1-4 exogenous processes and converges after at most three online interaction iterations (fig. 6b). Once trained, it can solve nearly all tasks in `Coffee`, `Domino`, and `Fan`, and over 80% in `Boil` and `Grow`. In comparison, we find that **MAPLE** is unable to achieve a high level of success even with 1000 times more interactions and evaluated only on the training distribution. We hypothesize that this is largely due to the challenge of exploration with sparse rewards. **ViLa** (zero-shot) was not able to do well in any domains. The few-shot variant achieves good performance in simple domains where satisficing plans share significant similarity with the demonstration plan (e.g., in `Coffee`, one simply needs to perform `Pour` and `NoOp` more times than in the demonstration). We observe that its performance degrades significantly in other domains where it must identify additional rules through trial and error (e.g., the requirement of matching colors in `Grow`) or domains that require compositional generalization (sequencing skills in potentially new ways). **VisPred** also struggles in these tasks because it learns

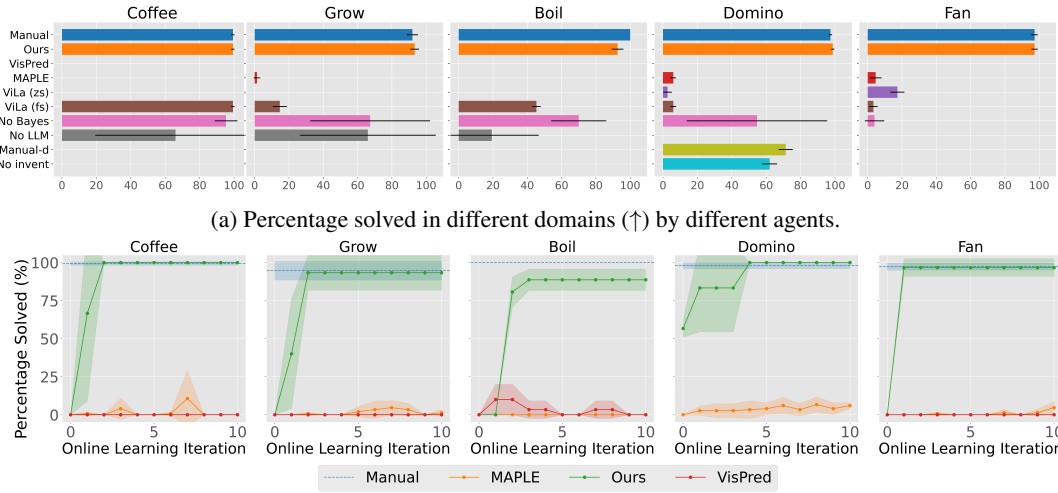

(a) Percentage solved in different domains (↑) by different agents.

(b) Learning curves of different online learning agents.

Figure 6: Performance metrics for various agents across different domains. The error bars/shaded regions show $\pm$ 1 standard deviation.

in a highly constrained model space. It attempted to learn the exogenous processes as different operators for the `NoOp` skill, but fell short due to an overly strong inductive bias. This bias restricts preconditions to only include atoms with variables already present in an operator's effects or option, which is especially limiting for the `NoOp` skill, where "robot" is the only variable. Moreover, it does not learn about the varying delays for different processes, a feature crucial for effective and efficient planning (e.g., in `Boil`). We note that ExoPredicator is unable to solve all tasks in `Boil`, partly because it failed to recognize the full disjunctive condition under which a spill can happen: it learned a process for water spilling when there is nothing under the faucet, but not when the jug underneath is full.

**(Q2). Ours** achieves the same (in `Coffee` and `Fan`) or better (in `Grow` and `Domino`) performance as **Manual** which uses manually engineered abstractions. We attribute this to the parameters learned via variational inference instead of being manually tuned. Furthermore, the near-zero performance of **No invent** and **Manual-d** underscores the importance of model learning and parameter learning. For example, in `Domino` their incomplete knowledge of the environment's dynamics and delay caused them to classify most tasks as unsolvable.

**(Q3).** Both the Bayesian model learning and the LLM guidance play a critical role in efficient, effective, and robust model structure learning. Without LLM guidance, the size of the search space for each process becomes astronomically large (sometimes reaching $2^{50}$), making it intractable to score all possible conditions. Without computing the Bayesian posterior of the data and model, the selection is based entirely on the prior in the LLM, which is not always reliable, especially with uncommon or unseen environment dynamics.

## 7 RELATED WORKS

**Temporal Planning.** Classical planners handle effects after a delay with durative actions (PDDL 2.1) (Fox & Long, 2003) and with autonomous processes and events (PDDL+) (Fox & Long, 2002); (RDDL) (Sanner et al., 2010), while heuristic search–based temporal planners such as COLIN (Coles et al., 2012) or OPTIC (Benton et al., 2012) support numeric fluents and deadlines. These works assume the full domain description is given. Our contribution is complementary: we *learn* models—including conditional stochastic delays—directly from a small number of trajectories.

**Hierarchical Reinforcement Learning.** HRL uses temporally extended actions to address long-horizon decision-making (Barto & Mahadevan, 2003). While many skill-learning approaches exist, they typically adopt the (semi-)Markov assumption at the option level: the distribution over outcomes depends only on the initiation state and the chosen option (Masson et al., 2016; Nasiriany

et al., 2022; Mishra et al., 2023). This fails to explicitly model exogenous dynamics or variable delays, attributing all changes to the agent's actions. We relax this assumption by learning a model for these external dynamics and stochastic delays, allowing a fixed set of skills to be flexibly used as the environment evolves.

**Hierarchical Reinforcement Learning.** HRL uses temporally extended actions to address long-horizon decision-making (Barto & Mahadevan, 2003). While many skill-learning approaches exist, they typically adopt the (semi-)Markov assumption at the option level: the distribution over outcomes depends only on the initiation state and the chosen option (Masson et al., 2016; Nasiriany et al., 2022; Mishra et al., 2023). In such formulations, environment-driven or exogenous dynamics can be absorbed into the option's transition model, but are not usually represented as separate causal processes with their own activation conditions and delay distributions. Our approach is complementary: we learn an explicit model for these external dynamics and stochastic delays, allowing a fixed set of skills to be used more flexibly as the environment evolves and enabling more generalizable planning.

**Large Foundation Models for Robotics.** Approaches such as SayCan (Ahn et al., 2022), RT-2 (Brohan et al., 2023), Inner Monologue (Huang et al., 2022), Code-as-Policy (Liang et al., 2023), ViLA (Hu et al., 2023), and $\pi_0$ (Black et al., 2025) treat planning as prompting: a pretrained LLM/VLM selects or synthesizes the next action at each step. These approaches inherit strong general-language priors, but—because they do not learn a world model—struggle to reason about concurrent processes (e.g. water keeps heating) or about actions whose effects materialize only if certain conditions persist. Our method calls foundation models for predicate invention (as in VisualPredicator) and model learning, yet it grounds their suggestions in experience and learns symbolic world models that supports look-ahead search.

**Causal Reasoning and Causal RL.** Structural causal models (SCMs) (Pearl, 2009) and their dynamic extensions form the foundation of various recent causal-RL algorithms (Buesing et al., 2019; Hammond et al., 2023; Zeng et al., 2025). These approaches assume that the underlying causal graph is either known or learnable at the feature level, but they do not tackle the challenges of symbolic abstraction or planning over durative processes. In contrast, ExoPredicator learns a causally consistent SCM (Rubenstein et al., 2017) whose variables are invented predicates and whose mechanisms are the learned causal processes, which enables reasoning at a higher level of abstraction.

**Learning Abstractions for Planning.** Early work learned STRIPS or NDR transition rules from demonstrations given a fixed predicate set (Pasula et al., 2007; Silver et al., 2021; 2022; Chitnis et al., 2022). Recent methods invent new predicates to improve generalisation (Silver et al., 2023; Liang et al., 2024); however, they typically abstract dynamics through agent-centric actions, without explicitly modeling exogenous causal processes that unfold over time in the background,. ExoPredicator extends predicate-invention to environments with exogenous dynamics and delayed causal effect.

## 8    CONCLUSION

We presented ExoPredicator, an integrated approach for learning and planning with causal processes in environments with exogenous dynamics and delayed effects. Our method demonstrates the ability to learn abstract world models from limited data, generalizing to new tasks with unseen objects and goals across various simulated environments, and outperforming key baselines. Future work will scale the framework to more complex, noisier, larger-scale environments, enhance learning with foundation models, and explore the interplay between skill and world modeling.

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

CONTENTS

# A    ADDITIONAL APPROACH DETAILS

## A.1    CAUSAL PROCESS SEMANTICS

We formalize the semantics of our causal process model, which underpins the planner described in section 4. We begin by defining a **small-step transition function** that describes the world's evolution at the finest temporal granularity, advancing one discrete timestep at a time. This detailed model allows us to precisely specify how and when processes are activated and their effects are applied. We then build upon this to define the **big-step transition function**, $T_{\text{big}}$, which abstracts away these fine-grained details. This function enables the planner to efficiently jump between significant changes in the abstract state, which is crucial for tractable long-horizon planning.

**Small-Step Semantics.** We model the world's evolution in discrete timesteps $t \in \mathbb{N}$. A complete snapshot of the world, or the **world state**, is a tuple $w_t = \langle s_t, Q_t, H_t \rangle$, where:

- $s_t$ is the set of ground atoms that are currently true.

- $Q_t$ is the *event dictionary*, a dictionary of scheduled effects of the form $\langle \underline{L}, t_{\text{start}} \rangle$, keyed by their end time $t_{\text{end}}$.

- $H_t$ is the history of all past atomic states, $[s_0, s_1, \ldots, s_{t-1}]$.

The world's fundamental dynamics are defined by a **small-step transition function**, $\mathcal{T}_{\text{small}}(w_t, \alpha_t) \mapsto w_{t+1}$, which advances the world by a single timestep. The transition given a potential agent command $\alpha_t$ (which may be an ground endogenous process or `None`) occurs in three stages:

1. **Event Execution:** Effects from events due at time $t$ are applied. We initialize $s_{t+1} \leftarrow s_t$. For every event $\langle \underline{L}, t_{\text{start}} \rangle$ in $Q_t$ scheduled for time $t$, if its overall condition $O_{\underline{L}}$ held from the step after activation up to the previous step, (i.e., for all $s_i \in H_t$ where $i > t_{\text{start}}$), its effects are applied: $s_{t+1} \leftarrow (s_{t+1} \setminus E_{\underline{L}}.\text{Del}) \cup E_{\underline{L}}.\text{Add}$.

2. **Process Activation:** New events are scheduled based on the state $s_{t+1}$ and the agent's command.
   - *Endogenous Activation:* If the agent issues a command $\alpha_t = \underline{L}_{\text{en}}$ and the process's start condition $C_{\text{start},\underline{L}_{\text{en}}}$ is satisfied in $s_{t+1}$, a delay $d \sim p_{\underline{L}_{\text{en}}}^{\text{delay}}$ is sampled (with $d \geq 1$). A new event $\langle \underline{L}_{\text{en}}, t \rangle$ is added to the queue for time $t + d$.
   - *Exogenous Activation:* For every exogenous process $\underline{L}_{\text{ex}}$, if its start condition $C_{\text{start},\underline{L}_{\text{ex}}}$ is satisfied in $s_{t+1}$ but was *not* satisfied in the previous state $s_{t-1}$ (i.e., it is edge-triggered), a delay $d \sim p_{\underline{L}_{\text{ex}}}^{\text{delay}}$ is sampled. A new event $\langle \underline{L}_{\text{ex}}, t \rangle$ is added to the queue for time $t + d$.

3. **State Finalization:** The next world state is $w_{t+1} = \langle s_{t+1}, Q'_t, H_t \cup \{s_t\} \rangle$, where $Q'_t$ is the updated event dictionary.

**Big-Step Semantics.** We define a **big-step transition function**, $\mathcal{T}_{\text{big}}(w_t, \underline{L}_{\text{en}})$, which computes the resulting world state after executing a single ground endogenous process $\underline{L}_{\text{en}}$ starting from world state $w_t$, or after simply waiting for the world to change ($\underline{L}_{\text{en}} = \text{NoOp}$).

This function simulates the environment forward by applying the small-step transition function $\mathcal{T}_{\text{small}}$ iteratively. The simulation proceeds until the chosen action $\underline{L}_{\text{en}}$ has completed or a maximum horizon $K_{\text{max}}$ is reached.

The transition $\mathcal{T}_{\text{big}}(w_t, \underline{L}_{\text{en}}) \mapsto w_{t+k}$ is computed as follows:

1. **Initialization:**
   - Initialize a step counter: $k \leftarrow 0$.
   - Set the initial world state for the simulation: $w'_k \leftarrow w_t$.
   - The endogenous process $\underline{L}_{\text{en}}$ is set as the command for the first step. Let the command for step $i$ be denoted $\alpha_i$. So, $\alpha_k \leftarrow \underline{L}_{\text{en}}$. For all subsequent steps $i > 0$, the command is null: $\alpha_i \leftarrow \text{None}$.

2. **Simulation Loop:** While the action $\underline{L}_{\text{en}}$ is still considered active and $k < K_{\text{max}}$:
   - Apply the small-step transition: $w'_{k+1} \leftarrow \mathcal{T}_{\text{small}}(w'_k, \alpha_k)$.
   - Increment the step counter: $k \leftarrow k + 1$.

- The action $\underline{L}_{\text{en}}$ is considered complete if its corresponding event has been executed within the simulation. This is tracked implicitly by the simulator state. A special case is the `NoOp` action, which is considered complete if any atom changes in the state, allowing the agent to wait for exogenous events.

3. **Final State:** The resulting world state is the state at the end of the simulation loop: $w_{t+k} \leftarrow w'_k$.

## A.2 FAST FORWARD HEURISTIC FOR CAUSAL PROCESSES

The Fast-Forward (FF) heuristic (Hoffmann, 2001) is a domain-independent planning heuristic that estimates the distance to goal by solving a relaxed version of the planning problem. We adapt this heuristic to our causal process framework, accounting for both endogenous and exogenous processes, as well as derived predicates.

**Relaxed Planning Graph Construction**   The FF heuristic constructs a Relaxed Planning Graph (RPG) by iteratively applying all applicable processes without considering delete effects. Given a state with atoms $s$, the heuristic proceeds as follows:

1. **Initialization**: Start with the current atoms $s$, augmented with any derived predicates that hold given those atoms.

2. **Forward Propagation**: For each layer $i$:
   - Find all processes whose $C$ (condition at start) is satisfied by facts in layer $i-1$
   - Add all add effects $E$.Add from these processes to create layer $i$ (ignoring delete effects $E$.Del)
   - Incrementally compute new derived predicates based on newly added primitive facts

3. **Termination**: Stop when the goal atoms $g \subseteq$ layer $i$, or when a fixed point is reached (no new facts can be added).

**Incremental Derived Predicate Computation**   To efficiently handle derived predicates, we maintain a dependency map from auxiliary predicates to derived predicates. When new primitive facts are added to a layer, we:

1. Identify which derived predicates might be affected based on their auxiliary predicate dependencies

2. Incrementally evaluate only those derived predicates on the updated state

3. Propagate newly derived facts through the dependency chain until a fixed point is reached

This avoids redundant recomputation of derived predicates that cannot be affected by the new facts.

**Relaxed Plan Extraction**   Once the RPG is built, we extract a relaxed plan via backward search:

1. Start with the goal atoms as subgoals to achieve

2. For each layer $i$ from $n$ to 1:
   - For each subgoal appearing for the first time in layer $i$:
     - If it's a derived predicate, replace it with its supporting auxiliary predicates
     - If it's a primitive predicate, find a process from layer $i-1$ that achieves it
   - Add the preconditions of selected processes as new subgoals
   - Count only endogenous processes toward the heuristic value

**Heuristic Value**   The heuristic value $h_{\text{FF}}(s)$ is the number of endogenous processes in the extracted relaxed plan. Exogenous processes are treated as having zero cost, reflecting that they occur automatically when their conditions are met. Formally:

$$h_{\text{FF}}(s) = |\{L \in \text{RelaxedPlan} : L \text{ is endogenous}\}| \qquad (1)$$

This provides an admissible estimate when all action costs are uniform, and guides the search toward states that require fewer agent interventions to reach the goal.

**Implementation Notes**   The implementation uses several optimizations:

- **Add-effect indexing**: We maintain a map from atoms to processes that add them, enabling efficient backward search during plan extraction

- **Early termination**: If the RPG reaches a fixed point without achieving the goal, we return $h = \infty$

- **Zero-cost exogenous processes**: These are included in the RPG construction but not counted in the final heuristic value, allowing the planner to leverage environmental dynamics

### A.3   PROBABILISTIC MODEL AND ELBO DERIVATION

The derivation for the probabilistic model is:

$p(\{s_t\}, \{\Delta_t^{\underline{L}}\})$
= (the law of conditional probability)

$$\prod_{t=1}^{T} p\left(s_t, \{\Delta_t^{\underline{L}}\}|s_{1:t-1}, \{\Delta_{1:t-1}^{\underline{L}}\}\right)$$

= (the law of conditional probability)

$$\prod_{t=1}^{T} p\left(\{\Delta_t^{\underline{L}}\}|s_{1:t}, \{\Delta_{1:t-1}^{\underline{L}}\}\right) \times p\left(s_t|s_{1:t-1}, \{\Delta_{1:t-1}^{\underline{L}}\}\right)$$

= (conditional independence of the delay vars.)

$$\left(\prod_{t=1}^{T} \prod_{\underline{L}} p\left(\Delta_t^{\underline{L}}|s_{1:t}\right)\right) \times \left(\prod_{t=1}^{T} p\left(s_t|s_{1:t-1}, \{\Delta_{1:t-1}^{\underline{L}}\}\right)\right)$$

= (by the structure of our model)

$$\left(\prod_{t=1}^{T} \prod_{\underline{L}} p_{\underline{L}}^{\text{delay}}(\Delta_t^{\underline{L}})^{C_{\underline{L}}(s_{1:t})}\right) \times \left(\prod_{t=1}^{T} \frac{1}{Z_t} F(s_t|s_{t-1}) \prod_{\underline{L}} \prod_{t'=1}^{t-1} E_{\underline{L}}(s_t)^{C_{\underline{L}}(s_{1:t'})O_{\underline{L}}(s_{t'+1:t-1})1[t=t'+\Delta_{t'}^{\underline{L}}]}\right)$$

where $Z_t = \sum_{s_t \in \mathcal{S}} F(s_t|s_{t-1}) \prod_{\underline{L}} \prod_{t'=1}^{t-1} E_{\underline{L}}(s_t)^{C_{\underline{L}}(s_{1:t'})O_{\underline{L}}(s_{t'+1:t-1})1[t=t'+\Delta_{t'}^{\underline{L}}]}$

= (assume factored state $s_t = (s_t^1, s_t^2, ...)$)

$$\left(\prod_{t=1}^{T} \prod_{\underline{L}} p_{\underline{L}}^{\text{delay}}(\Delta_t^{\underline{L}})^{C_{\underline{L}}(s_{1:t})}\right) \times \left(\prod_{t=1}^{T} \prod_{j=1}^{J} \frac{1}{Z_t^j} F(s_t^j|s_{t-1}^j) \prod_{\underline{L}} \prod_{t'=1}^{t-1} E_{\underline{L}}^j(s_t^j)^{C_{\underline{L}}(s_{1:t'})O_{\underline{L}}(s_{t'+1:t-1})1[t=t'+\Delta_{t'}^{\underline{L}}]}\right)$$

where $Z_t^j = \sum_{s_t^j \in \mathcal{S}^j} F(s_t^j|s_{t-1}^j) \prod_{\underline{L}} \prod_{t'=1}^{t-1} E_{\underline{L}}^j(s_t^j)^{C_{\underline{L}}(s_{1:t'})O_{\underline{L}}(s_{t'+1:t-1})1[t=t'+\Delta_{t'}^{\underline{L}}]}$

With the change of basis, our model becomes:

$p(\{s_t\}, \{A_t^{\underline{L}}\}) =$

$$\left(\prod_{t=1}^{T} \prod_{\underline{L}} p_{\underline{L}}^{\text{delay}}(A_t^{\underline{L}} - t)^{C_{\underline{L}}(s_{1:t})}\right) \times \left(\prod_{t=1}^{T} \prod_{j=1}^{J} \frac{1}{Z_t^j} F(s_t^j|s_{t-1}^j) \prod_{\underline{L}} \prod_{t'=1}^{t-1} E_{\underline{L}}^j(s_t^j)^{C_{\underline{L}}(s_{1:t'})O_{\underline{L}}(s_{t'+1:t-1})1[A_{t'}^{\underline{L}}=t]}\right)$$

where $Z_t^j = \sum_{\hat{s}_t^j \in \mathcal{S}^j} F(\hat{s}_t^j|s_{t-1}^j) \prod_{\underline{L}} \prod_{t'=1}^{t-1} E_{\underline{L}}^j(\hat{s}_t^j)^{C_{\underline{L}}(s_{1:t'})O_{\underline{L}}(s_{t'+1:t-1})1[A_{t'}^{\underline{L}}=t]}$

The derivation for the ELBO is:

$$\log p\left(\{s_t\}\right)$$

$$= \log \int \frac{q(\{A_t^L\})}{q(\{A_t^L\})} p(\{s_t\}, \{A_t^L\}) d\{A_t^L\}$$

$$= \log \mathbb{E}_q \left[ \frac{p(\{s_t\}, \{A_t^L\})}{q(\{A_t^L\})} \right]$$

$$\geq \mathbb{E}_q \left[ \log \frac{p(\{s_t\}, \{A_t^L\})}{q(\{A_t^L\})} \right]$$

$$= \mathbb{E}_q \left[ \log p(\{s_t\}, \{A_t^L\})) \right] + \mathbb{H}_q \left[ \{A_t^L\} \right]$$

$$= \sum_{t=1}^{T} \sum_{\underline{L}} \underbrace{\mathbb{E}_q \left[ C_{\underline{L}}(s_{1:t}) \log \left( p_{\underline{L}}^{\text{delay}}(A_t^L - t) \right) \right]}_{\text{expected log delay probability}} +$$

$$\sum_{t=1}^{T} \sum_{j=1}^{J} \log \left( F(s_t^j | s_{t-1}^j) \right) + \sum_{\underline{L}} \sum_{t'=1}^{t-1} \underbrace{\mathbb{E}_q \left[ C_{\underline{L}}(s_{1:t'}) O_{\underline{L}}(s_{t'+1:t-1}) 1[A_{t'}^L = t] \log \left( E_{\underline{L}}^j(s_t^j) \right) \right]}_{\text{unnormalized expected log state probability}} -$$

$$\sum_{t=1}^{T} \sum_{j=1}^{J} \mathbb{E}_q \left[ \log \left( \underbrace{\sum_{\hat{s}_t^j \in \mathcal{S}^j} F(\hat{s}_t^j | s_{t-1}^j) \prod_{\underline{L}} \prod_{t'=1}^{t-1} E_{\underline{L}}^j(\hat{s}_t^j)^{C_{\underline{L}}(s_{1:t'}) O_{\underline{L}}(s_{t'+1:t-1}) 1[A_{t'}^L=t]}}_{Z_t^j} \right) \right] + \mathbb{H}_q \left[ \{A_t^L\} \right]$$

$$\geq \text{(reverse bound the expected normalization constant; independence of } A_t^L)$$

$$\sum_{t=1}^{T} \sum_{\underline{L}} \mathbb{E}_q \left[ C_{\underline{L}}(s_{1:t}) \log \left( p_{\underline{L}}^{\text{delay}}(A_t^L - t) \right) \right] +$$

$$\sum_{t=1}^{T} \sum_{j=1}^{J} \log \left( F(s_t^j | s_{t-1}^j) \right) + \sum_{\underline{L}} \sum_{t'=1}^{t-1} \mathbb{E}_q \left[ C_{\underline{L}}(s_{1:t'}) O_{\underline{L}}(s_{t'+1:t-1}) 1[A_{t'}^L = t] \log \left( E_{\underline{L}}^j(s_t^j) \right) \right] -$$

$$\sum_{t=1}^{T} \sum_{j=1}^{J} \log \sum_{\hat{s}_t^j \in \mathcal{S}^j} F(\hat{s}_t^j | s_{t-1}^j) \prod_{\underline{L}} \prod_{t'=1}^{t-1} \mathbb{E}_q \left[ E_{\underline{L}}^j(\hat{s}_t^j)^{C_{\underline{L}}(s_{1:t'}) O_{\underline{L}}(s_{t'+1:t-1}) 1[A_{t'}^L=t]} \right] + \mathbb{H}_q \left[ \{A_t^L\} \right]$$

$$= \text{(expand the expectation out independently)}$$

$$\sum_{t=1}^{T} \sum_{\underline{L}} C_{\underline{L}}(s_{1:t}) \sum_{A_t^L=t+1}^{T} q_t^L(A_t^L) \log \left( p_{\underline{L}}^{\text{delay}}(A_t^L - t) \right) +$$

$$\sum_{t=1}^{T} \sum_{j=1}^{J} \log \left( F(s_t^j | s_{t-1}^j) \right) + \sum_{\underline{L}} \sum_{t'=1}^{t-1} q_{t'}^L(A_{t'}^L = t) C_{\underline{L}}(s_{1:t'}) O_{\underline{L}}(s_{t'+1:t-1}) \log \left( E_{\underline{L}}^j(s_t^j) \right) -$$

$$\sum_{t=1}^{T} \sum_{j=1}^{J} \log \sum_{\hat{s}_t^j \in \mathcal{S}^j} F(\hat{s}_t^j | s_{t-1}^j) \prod_{\underline{L}} \prod_{t'=1}^{t-1} \left( q_{t'}^L(A_{t'}^L = t) \underbrace{E_{\underline{L}}^j(\hat{s}_t^j)^{O_{\underline{L}}(s_{t'+1:t-1})}}_{\exp\left(O_{\underline{L}}(s_{t'+1:t-1}) \log E_{\underline{L}}^j(\hat{s}_t^j)\right)} + \left( 1 - q_{t'}^L(A_{t'}^L = t) \right) \right)^{C_{\underline{L}}(s_{1:t'})} +$$

$$\mathbb{H}_q \left[ \{A_t^L\} \right]$$

Evaluating this objective is computationally intensive due to the nested loops, but can be simplified by some algebraic refactoring. In the first term, the second sum only needs to loop through the non-zero terms, which are laws whose conditions are satisfied at time $t$.

In the second term, the sum over $j$ and $L$ can be reduced to a sum over $s_t^j$s that are in the effects of some laws, and thus have non-zero $\log\left(E_{\underline{L}}^j(s_t^j)\right)$ plus the log frame strength from unchanged atoms, and the sum over $t'$ can be reduced to just steps where the law is activated, similar to the first term above.

For experiments, we use a categorical variational distribution over the discrete support of the delays. The variational parameters are initialized to be uniform over this support.

### A.4 PROCESS LEARNING DETAILS

Given a set of trajectories $\mathcal{D}_{\mathrm{abs}}$, a set of predicates $\Psi$, and the agent's known endogenous processes $\mathcal{L}_{en}$, we learn the set of exogenous processes $\mathcal{L}_{ex}$. Our method follows Chitnis et al. (2022) in assuming that for any given effect (a unique pair of add/delete atoms), there is at most one exogenous process that causes it. While this prevents learning multiple distinct causes for the same outcome, it significantly simplifies the search problem. Any lost expressivity can be recovered by inventing more nuanced predicates. The learning algorithm proceeds in five steps:

1. **Segment.** First, we split each raw trajectory into shorter segments based on changes in the abstract state. Specifically, a new segment begins whenever the set of true predicates changes. Each segment therefore contains a sequence of constant abstract states followed by a single timestamp where the state changes.

2. **Filter.** Next, we filter out any segments where the observed state change can be explained by one of the agent's known endogenous processes. For example, if the agent executes the `Pick` action and the `Holding` predicate becomes true, that segment is attributed to an endogenous process and removed from consideration. This ensures we only attempt to learn models for effects caused by the environment's own dynamics.

3. **Cluster.** We then cluster the remaining segments based on their effects. We assume that each exogenous process has a single, atomic effect (e.g., one predicate changing from false to true).[2] If a segment involves multiple predicate changes, we duplicate it into multiple clusters—one for each change—allowing us to learn a separate process for each atomic effect. This induces a deterministic partition of segments by effect, so the step introduces no clustering hyperparameters.

4. **Intersect.** For each cluster, we identify a set of potential preconditions for the associated effect. This is done by taking the set intersection of all predicates that were true at the start of every segment in the cluster. This step produces a superset of candidate atoms for the process's conditions.

5. **Select.** The intersection from the previous step often contains many irrelevant atoms. To find the true preconditions, we first use an LLM to propose a small number of plausible condition sets from this large superset (the prompt is detailed in the end of this section). We then use Bayesian model selection to score each candidate condition set $C_i$ and select the one that maximizes the posterior probability:

$$L_{C^*,E} = \arg\max_{L_{C_i,E}} \log p(L_{C_i,E}|\mathcal{D}_{\mathrm{abs}}) = \arg\max_{L_{C_i,E}} \left(\log p(\mathcal{D}_{\mathrm{abs}}|L_{C_i,E}) + \log p(L_{C_i,E})\right)$$

where $\log p(\mathcal{D}_{\mathrm{abs}}|L_{C_i,E})$ is the approximate marginal likelihood from the previous section and $\log p(L_{C_i,E})$ is a minimum description length prior that penalizes overly complex conditions.

Process condition proposal prompt

---

[2] This imposes no loss of generality, because separate processes can be learned for each changed predicate.

```
You are an expert in automated planning and causal reasoning. Your
    task is to propose the most likely sets of conditions for
    specific process effects to occur.

Given a process with specific add effects and delete effects, you
    need to propose multiple coherent sets of candidate atoms that
    could serve as necessary conditions for the process to
    successfully achieve its effects.

Key principles for proposing condition sets:
1. **Causal Relevance**: Each set should contain atoms that are
    causally necessary for the effects to occur
2. **Physical Constraints**: Include atoms representing physical
    constraints or requirements
3. **Domain Knowledge**: Use common sense about how processes work
    in the real world
4. **State Dependencies**: Include atoms that represent
    prerequisite states for the effects
5. Terminal-Progress Exclusion: If an add effect is a
    terminal/complete state within a progression family, do not
    include any intermediate/progress predicates from the same
    family as preconditions (e.g., Partially*, Started, InProgress,
    HasSome).

Available predicates in the candidate atoms are:
{PREDICATE_LISTING}

For each process, I will provide:
- Add effects: What the process makes true
- Delete effects: What the process makes false
- Candidate atoms: Potential precondition atoms to choose from

{PROCESS_EFFECTS_AND_CANDIDATES}

Please propose as many likely condition sets as you deem suitable
    for each process. Each condition set should be a coherent
    combination of atom indices that together form a plausible set
    of preconditions. It's possible that there is a large number of
    atoms in a condition set in some cases.

Think step by step if it's helpful before outputting your final
    response, formatted strictly as:
<answer>
Process 0:
Set 1: [2, 0, 4]
Set 2: [1, 3, 0, 5]
Set 3: [2, 4]
Process 1:
Set 1: [1, 3]
Set 2: [0, 1, 3]
...
</answer>
```

## A.5 PREDICATE LEARNING DETAILS

Our approach to predicate learning follows the general methodology of prior work (Liang et al., 2024; Silver et al., 2023), where a foundation model (Gemini 2.5 Pro) is prompted to synthesize a set of candidate predicates adhering to a predefined API. The final subset of predicates is then selected by maximizing an approximate planning metric; a subset receives a higher score if it enables the planner to find plans that are similar to successful demonstrations while requiring fewer planning resources.

Our proposal process involves two stages. First, we prompt a VLM with a trajectory from the environment to propose a set of high-level, symbolic concepts that could be useful for planning. We use different prompts depending on whether the provided trajectory was successful or resulted in failure. In the second stage, these concepts are translated into executable Python code that matches our predicate object API.

Since the primary focus of this work is on learning abstract models in a more expressive and complex model space, we simplify the perception problem. We assume the agent has access to a state representation containing all the necessary object features to evaluate any relevant predicate, without needing to ground them directly in image data.

In our experiments, we found it sufficient to generate a pool of candidate predicates only once, based on the initial demonstration trajectories. This set of candidates is then retained and made available for the agent to select from during all subsequent online learning iterations. The full prompt templates used for predicate invention are provided below.

---

**Predicate invention from successful trajectory prompt template**

```
Context: You are an expert AI planning researcher. Your task is to
    design task-specific predicates that can be used in a PDDL-like
    model to facilitate effective and efficient robot planning.

### Types and Features
The environment has the following types, each with some features:

{TYPES_IN_ENV}

### Existing Predicates
You should consider the following existing predicates:

{PREDICATES_IN_ENV}

### Robot's Goal
The robot's ultimate goal in this environment is to make the
    predicate {GOAL_PREDICATE} true.

### Demonstration Trajectory
The demonstrator performs a sequence of actions, ending with the
    goal being achieved. The state-feature-action trajectory is
    provided below.

{EXPERIENCE_IN_ENV}

### Your Task
Invent a small set of the *most essential* new predicates. These
    should be simple, primitive concepts that represent critical
    **subgoals**, **conditions for subgoals**, or *any conditions
    that must be maintained to prevent failure, even if failure is
    not shown in the demonstration*.

Note:
- If a continuous feature represents progress toward a subgoal,
    define exactly one terminal predicate for its end state (with a
    high threshold near the value observed when the goal is
    achieved) and ignore intermediate progress states.
- Geometry/affordance guardrail: Do not invent predicates that
    rely on pose or derived geometry (distance, proximity,
    alignment, path, line-of-sight, "within theta", abs() < tau),
    or on capabilities (Can*, AbleTo*, Near*), unless such
    relations are already provided as explicit non-pose features.
```

```
- There maybe some risk or violation condition that must remain
    safe for the goal to succeed, even if failure is not shown. For
    such features include a maintenance predicate that keeps it
    within a safe bound (e.g., T_low <= 0.1 for normalized
    features).

### Constraints
- Do *not* propose any new predicates that are purely pose-based
    (i.e., based on raw x, y, z coordinates or 'rot', 'tilt'
    angles).
- Avoid composite predicates (no negation/AND/OR of other
    predicates). Each proposal should state one primitive property
    or relation that can be verified from the given features.
- Assertions must be clear and unambiguous, describing the
    relationship or properties of variables ?<var1>, ?<var2>, etc.,
    so an external observer could label truth values from the
    provided object features alone.
- Replace placeholders like <predicate1_name>, <var1>, etc., with
    actual names; <type1> and <obj1> with actual names and types
    from the state dictionary provided. *Do not* use types that are
    not present in the states (e.g., int or float).
- Do not use bold or italic fonts in your response.
- Respond only with the output section outlined below.

### Output Format
Provide your predicate proposals in the following format:

```plaintext
# Predicate Proposals
* <predicate1_name>(?<var1>:<type1>, ?<var2>:<type2>, ...): <The
    assertion this predicate is making>.
* ...
```
```

```
Predicate invention from failed trajectory prompt template

Context: You are an expert AI planning researcher. Your task is to
    design task-specific predicates that can be used in a PDDL-like
    model to (a) detect unreachable goals early and (b) avoid
    futile plans.

### Types and Features
The environment has the following types, each with some features:

{TYPES_IN_ENV}

### Existing Predicates
You should consider the following existing predicates:

{PREDICATES_IN_ENV}

### Robot's Goal
The robot's goal in this environment is to make the predicate
    {GOAL_PREDICATE} true.

### Demonstration Trajectory
The demonstrator performs a sequence of actions, which they
    thought would achieve the goal but did not. The
    state-feature-action trajectory is provided below.
```

```
{EXPERIENCE_IN_ENV}

### Your Task
Invent a small set of the most essential new predicates whose
    primary purpose is to expose **blocking conditions**:
    primitive, easily-checkable properties that must hold somewhere
    in the environment for the goal to be achievable. If a blocking
    condition is true (or a required enabling condition is false),
    a rational planner can conclude that the goal is unreachable
    under the available actions.

Prioritize:
- Necessary preconditions for success that the demonstrator
    overlooked.
- Irreversible or static properties (e.g., object class, material,
    color category) that make certain transitions impossible.
- Local checks that can be evaluated from the provided object
    features without simulating dynamics.

Note: Features such as colors may encode latent material or
    affordance classes. You may want to propose predicates that
    identify such classes when they impose hard constraints on
    feasible state transitions or interactions (e.g.,
    non-deformable, too-heavy, low-friction, brittle, sealed,
    non-activatable).
For every new predicate, explicitly state how its truth value is
    decided from the listed non-pose features. Use a deterministic
    rule with concrete feature names and numeric thresholds or
    categorical equalities (no vague phrases like "such as mass").
    If invoking a latent property, tie it to an explicit feature
    pattern.

### Constraints
- Geometry/pose prohibition (hard): Do not use or derive from pose
    fields (x, y, z, yaw, roll, tilt, wrist) or any geometric
    constructs (distance, proximity, alignment, vector, path,
    line-of-sight, "within theta", abs()<tau). Predicates
    mentioning these are invalid.
- Soft blacklist (names/definitions to avoid): aligned, path,
    near, distance, airstream, obstructed, adjacency (unless given
    as a non-pose feature).0
- Avoid composite predicates (no negation/AND/OR of other
    predicates). Each proposal should state one primitive property
    or relation that can be verified from the given features.
- Assertions must be clear and unambiguous, describing the
    relationship or properties of variables ?<var1>, ?<var2>, etc.,
    so an external observer could label truth values from the
    provided object features alone.
- Replace placeholders like <predicate1_name>, <var1>, etc., with
    actual names; use only types present in the states.
- Do not use bold or italic fonts.
- Respond only with the Output section below.

### Output Format
Provide your predicate proposals in the following format:

```plaintext
# Predicate Proposals
* <predicate1_name>(?<var1>:<type1>, ?<var2>:<type2>, ...): <The
    assertion this predicate is making>.
* ...
```

---

**Predicate implementation prompt template**

```
Context: You are an expert AI researcher tasked with inventing
    task-specific state abstraction predicates for effective and
    efficient robotic planning.

I will describe the API you should use for writing predicates and
    the environment the robot is in.
# The API for `Predicate` and `State` is:
{STRUCT_DEFINITION}

The environment includes the following object-type variables with
    features:
{TYPES_IN_ENV}
where `bbox_left`, `bbox_lower`, ..., corresponds to the pixel
    index of the left, lower boundary of the object bounding box in
    the image starting from (0, 0) at the bottom left corner of the
    image.
`pose_x`, `pose_y`, and `pose_z` correspond to the 3d object
    position in the world frame, so these are not comparable to the
    bbox values.

The existing predicates are:
{PREDICATES_IN_ENV}

The states the predicates have been evaluated on are:
{LISTED_STATES}

Please implement the following predicates which would have
    evaluation values that matches the following specification:

{PREDICATE_SPECS}

Implement each predicate in a seperate Python block as follows:
```python
def _<predicate_name>_holds(state: State, objects:
    Sequence[Object]) -> bool:
     # Implement the boolean classifier function here
     ...

# Define the predicate name here
name: str = ...

# A list of object-type variables for the predicate, using the
    ones defined in the environment
param_types: List[Type] = ...
<predicate_name> = Predicate(name, param_types, classifier)
```

- When writing the proposals, strictly adhere to the following
    guidlines:
     - Use only object-type variables defined in the environment
     when defining `param_types`.
     - Don't use any undefined constants;
     - Don't use object features that are not present in the
     definition of that object type.
     - Adhere to the type hints in the predicate definition
     template.
     - Make use of helper functions such as the classifier function
     in the existing predicates, if they're helpful.
     - Your don't need to import anything.
```

# B    ADDITIONAL ENVIRONMENT DETAILS

We describe the predicates and endogenous processes that we provide to ExoPredicator at the beginning of learning. In contrast, the baselines (**Manual**, **ViLa**, **MAPLE** and **VisualPredicator**) are provided with an expanded set of predicates that we intend our approach to discover autonomously.

## B.1    COFFEE

**Train/Test split**    The training tasks for this environment involve filling a single cup with coffee. The held-out test tasks require the agent to fill two or three cups. In both distributions, the size and color of the cups may vary.

**Goal predicates.**    {CupFilled}

**Initial predicates and endogenous processes.**    {JugAboveCup, OnTable, NotAboveCup, CupFilled, Holding, MachineOn, JugInMachine, HandEmpty}

```
EndogenousProcess-NoOp:
    Parameters: [?robot:robot]
    Conditions at start: []
    Conditions overall: []
    Conditions at end: []
    Add Effects: []
    Delete Effects: []
    Ignore Effects: [JugAboveCup, NotAboveCup]
    Log Strength: 1.0000
    Delay Distribution: ConstantDelay(1.0000)
    Option Spec: NoOp(?robot:robot),
```

```
EndogenousProcess-PickJugFromMachine:
    Parameters: [?robot:robot, ?jug:jug, ?machine:coffee_machine]
    Conditions at start: [HandEmpty(?robot:robot), JugInMachine(?jug:jug,
     ?machine:coffee_machine)]
    Conditions overall: []
    Conditions at end: []
    Add Effects: [Holding(?robot:robot, ?jug:jug)]
    Delete Effects: [HandEmpty(?robot:robot), JugInMachine(?jug:jug, ?
    machine:coffee_machine)]
    Ignore Effects: []
    Log Strength: 1.0000
    Delay Distribution: DiscreteGaussianDelay(3.0000, 0.1000)
    Option Spec: PickJug(?robot:robot, ?jug:jug),
```

```
EndogenousProcess-PickJugFromTable:
    Parameters: [?robot:robot, ?jug:jug]
    Conditions at start: [HandEmpty(?robot:robot), OnTable(?jug:jug)]
    Conditions overall: []
    Conditions at end: []
    Add Effects: [Holding(?robot:robot, ?jug:jug)]
    Delete Effects: [HandEmpty(?robot:robot), OnTable(?jug:jug)]
    Ignore Effects: []
    Log Strength: 1.0000
    Delay Distribution: DiscreteGaussianDelay(3.0000, 0.1000)
    Option Spec: PickJug(?robot:robot, ?jug:jug),
```

```
EndogenousProcess-PlaceJugInMachine:
    Parameters: [?robot:robot, ?jug:jug, ?machine:coffee_machine]
    Conditions at start: [Holding(?robot:robot, ?jug:jug), NotAboveCup(?
    robot:robot, ?jug:jug)]
    Conditions overall: []
    Conditions at end: []
```

```
    Add Effects: [HandEmpty(?robot:robot), JugInMachine(?jug:jug, ?
    machine:coffee_machine)]
    Delete Effects: [Holding(?robot:robot, ?jug:jug)]
    Ignore Effects: []
    Log Strength: 1.0000
    Delay Distribution: DiscreteGaussianDelay(4.0000, 0.1000)
    Option Spec: PlaceJugInMachine(?robot:robot, ?jug:jug, ?machine:
    coffee_machine),
```

```
EndogenousProcess-PourFromCup:
    Parameters: [?robot:robot, ?jug:jug, ?to_cup:cup, ?from_cup:cup]
    Conditions at start: [Holding(?robot:robot, ?jug:jug), JugAboveCup(?
    jug:jug, ?from_cup:cup)]
    Conditions overall: []
    Conditions at end: []
    Add Effects: [JugAboveCup(?jug:jug, ?to_cup:cup)]
    Delete Effects: [JugAboveCup(?jug:jug, ?from_cup:cup), NotAboveCup(?
    robot:robot, ?jug:jug)]
    Ignore Effects: [JugAboveCup, NotAboveCup]
    Log Strength: 1.0000
    Delay Distribution: DiscreteGaussianDelay(2.0000, 0.1000)
    Option Spec: Pour(?robot:robot, ?jug:jug, ?to_cup:cup),
```

```
EndogenousProcess-PourFromNotAboveCup:
    Parameters: [?robot:robot, ?jug:jug, ?cup:cup]
    Conditions at start: [Holding(?robot:robot, ?jug:jug), NotAboveCup(?
    robot:robot, ?jug:jug)]
    Conditions overall: []
    Conditions at end: []
    Add Effects: [JugAboveCup(?jug:jug, ?cup:cup)]
    Delete Effects: [NotAboveCup(?robot:robot, ?jug:jug)]
    Ignore Effects: []
    Log Strength: 1.0000
    Delay Distribution: DiscreteGaussianDelay(2.0000, 0.1000)
    Option Spec: Pour(?robot:robot, ?jug:jug, ?cup:cup),
```

```
EndogenousProcess-TurnMachineOn:
    Parameters: [?robot:robot, ?jug:jug, ?machine:coffee_machine]
    Conditions at start: [HandEmpty(?robot:robot), JugInMachine(?jug:jug,
     ?machine:coffee_machine)]
    Conditions overall: []
    Conditions at end: []
    Add Effects: [MachineOn(?machine:coffee_machine)]
    Delete Effects: []
    Ignore Effects: []
    Log Strength: 1.0000
    Delay Distribution: DiscreteGaussianDelay(2.0000, 0.1000)
    Option Spec: TurnMachineOn(?robot:robot, ?machine:coffee_machine)
```

**Additional predicates.**  {JugFilled}

## B.2  GROW

**Train/Test split.**    In the training tasks, the agent must grow plants in two pots. For each pot, at least one jug of a matching color is available, with a maximum of two jugs present in the environment overall. The test tasks increase in complexity, requiring the agent to grow plants in three pots, again with at least one matching jug available for each and a maximum of two jugs in total.

**Goal predicates.**    {Grown}

**Initial predicates and endogenous processes.** {NotAboveCup, JugOnTable, Holding}

```
EndogenousProcess-NoOp:
    Parameters: [?robot:robot]
    Conditions at start: []
    Conditions overall: []
    Conditions at end: []
    Add Effects: []
    Delete Effects: []
    Ignore Effects: []
    Log Strength: 1.0000
    Delay Distribution: DiscreteGaussianDelay(1.0000, 0.1000)
    Option Spec: NoOp(?robot:robot),
```

```
EndogenousProcess-PickJugFromTable:
    Parameters: [?robot:robot, ?jug:jug]
    Conditions at start: [HandEmpty(?robot:robot), JugOnTable(?jug:jug)]
    Conditions overall: []
    Conditions at end: []
    Add Effects: [Holding(?robot:robot, ?jug:jug)]
    Delete Effects: [HandEmpty(?robot:robot), JugOnTable(?jug:jug)]
    Ignore Effects: []
    Log Strength: 1.0000
    Delay Distribution: DiscreteGaussianDelay(3.0000, 0.1000)
    Option Spec: PickJug(?robot:robot, ?jug:jug),
```

```
EndogenousProcess-PlaceJugOnTable:
    Parameters: [?robot:robot, ?jug:jug, ?cup:cup]
    Conditions at start: [Holding(?robot:robot, ?jug:jug), JugAboveCup(?
    jug:jug, ?cup:cup)]
    Conditions overall: []
    Conditions at end: []
    Add Effects: [HandEmpty(?robot:robot), JugOnTable(?jug:jug),
    NotAboveCup(?robot:robot, ?jug:jug)]
    Delete Effects: [Holding(?robot:robot, ?jug:jug), JugAboveCup(?jug:
    jug, ?cup:cup)]
    Ignore Effects: [HandEmpty, Holding, JugAboveCup, JugOnTable,
    NotAboveCup]
    Log Strength: 1.0000
    Delay Distribution: DiscreteGaussianDelay(3.0000, 0.1000)
    Option Spec: Place(?robot:robot, ?jug:jug),
```

```
EndogenousProcess-PourFromAboveCup:
    Parameters: [?robot:robot, ?jug:jug, ?from_cup:cup, ?to_cup:cup]
    Conditions at start: [Holding(?robot:robot, ?jug:jug), JugAboveCup(?
    jug:jug, ?from_cup:cup)]
    Conditions overall: []
    Conditions at end: []
    Add Effects: [JugAboveCup(?jug:jug, ?to_cup:cup)]
    Delete Effects: [JugAboveCup(?jug:jug, ?from_cup:cup)]
    Ignore Effects: [JugAboveCup, NotAboveCup]
    Log Strength: 1.0000
    Delay Distribution: DiscreteGaussianDelay(2.0000, 0.1000)
    Option Spec: Pour(?robot:robot, ?jug:jug, ?to_cup:cup),
```

```
EndogenousProcess-PourFromNotAboveCup:
    Parameters: [?robot:robot, ?jug:jug, ?cup:cup]
    Conditions at start: [Holding(?robot:robot, ?jug:jug), NotAboveCup(?
    robot:robot, ?jug:jug)]
    Conditions overall: []
    Conditions at end: []
    Add Effects: [JugAboveCup(?jug:jug, ?cup:cup)]
    Delete Effects: [NotAboveCup(?robot:robot, ?jug:jug)]
```

```
    Ignore Effects: [JugAboveCup, NotAboveCup]
    Log Strength: 1.0000
    Delay Distribution: DiscreteGaussianDelay(2.0000, 0.1000)
    Option Spec: Pour(?robot:robot, ?jug:jug, ?cup:cup)
```

**Additional predicates.** {SameColor}

## B.3 BOIL

**Train/Test split.** Training tasks require the agent to boil a single jug of water. The evaluation includes tasks that involve boiling either one or two jugs.

**Goal predicates.** {HumanHappy}

**Initial predicates and endogenous processes.** { FaucetOn, FaucetOff, HumanHappy, JugAtBurner, Holding, JugAtFaucet, NoJugAtBurner, BurnerOff, HandEmpty, BurnerOn, NoJugAtFaucet, JugNotAtBurnerOrFaucet}

```
EndogenousProcess-NoOp:
    Parameters: [?robot:robot]
    Conditions at start: []
    Conditions overall: []
    Conditions at end: []
    Add Effects: []
    Delete Effects: []
    Ignore Effects: []
    Log Strength: 1.0000
    Delay Distribution: ConstantDelay(1.0000)
    Option Spec: NoOp(?robot:robot),
```

```
EndogenousProcess-PickJugFromBurner:
    Parameters: [?robot:robot, ?jug:jug, ?burner:burner]
    Conditions at start: [HandEmpty(?robot:robot), JugAtBurner(?jug:jug,
    ?burner:burner)]
    Conditions overall: []
    Conditions at end: []
    Add Effects: [Holding(?robot:robot, ?jug:jug), NoJugAtBurner(?burner:
    burner)]
    Delete Effects: [HandEmpty(?robot:robot), JugAtBurner(?jug:jug, ?
    burner:burner)]
    Ignore Effects: []
    Log Strength: 1.0000
    Delay Distribution: DiscreteGaussianDelay(4.0000, 0.1000)
    Option Spec: PickJug(?robot:robot, ?jug:jug),
```

```
EndogenousProcess-PickJugFromFaucet:
    Parameters: [?robot:robot, ?jug:jug, ?faucet:faucet]
    Conditions at start: [HandEmpty(?robot:robot), JugAtFaucet(?jug:jug,
    ?faucet:faucet)]
    Conditions overall: []
    Conditions at end: []
    Add Effects: [Holding(?robot:robot, ?jug:jug), NoJugAtFaucet(?faucet:
    faucet)]
    Delete Effects: [HandEmpty(?robot:robot), JugAtFaucet(?jug:jug, ?
    faucet:faucet)]
    Ignore Effects: []
    Log Strength: 1.0000
    Delay Distribution: DiscreteGaussianDelay(4.0000, 0.1000)
    Option Spec: PickJug(?robot:robot, ?jug:jug),
```

```
EndogenousProcess-PickJugFromOutsideFaucetAndBurner:
    Parameters: [?robot:robot, ?jug:jug]
    Conditions at start: [HandEmpty(?robot:robot), JugNotAtBurnerOrFaucet
    (?jug:jug)]
    Conditions overall: []
    Conditions at end: []
    Add Effects: [Holding(?robot:robot, ?jug:jug)]
    Delete Effects: [HandEmpty(?robot:robot), JugNotAtBurnerOrFaucet(?jug
    :jug)]
    Ignore Effects: []
    Log Strength: 1.0000
    Delay Distribution: DiscreteGaussianDelay(3.0000, 0.1000)
    Option Spec: PickJug(?robot:robot, ?jug:jug),
```

```
EndogenousProcess-PlaceOnBurner:
    Parameters: [?robot:robot, ?jug:jug, ?burner:burner]
    Conditions at start: [Holding(?robot:robot, ?jug:jug), NoJugAtBurner
    (?burner:burner)]
    Conditions overall: []
    Conditions at end: []
    Add Effects: [HandEmpty(?robot:robot), JugAtBurner(?jug:jug, ?burner:
    burner)]
    Delete Effects: [Holding(?robot:robot, ?jug:jug), NoJugAtBurner(?
    burner:burner)]
    Ignore Effects: []
    Log Strength: 1.0000
    Delay Distribution: DiscreteGaussianDelay(5.0000, 0.1000)
    Option Spec: PlaceOnBurner(?robot:robot, ?burner:burner),
```

```
EndogenousProcess-PlaceOutsideFaucetAndBurner:
    Parameters: [?robot:robot, ?jug:jug]
    Conditions at start: [Holding(?robot:robot, ?jug:jug)]
    Conditions overall: []
    Conditions at end: []
    Add Effects: [HandEmpty(?robot:robot), JugNotAtBurnerOrFaucet(?jug:
    jug)]
    Delete Effects: [Holding(?robot:robot, ?jug:jug)]
    Ignore Effects: []
    Log Strength: 1.0000
    Delay Distribution: DiscreteGaussianDelay(3.0000, 0.1000)
    Option Spec: PlaceOutsideBurnerAndFaucet(?robot:robot),
```

```
EndogenousProcess-PlaceUnderFaucet:
    Parameters: [?robot:robot, ?jug:jug, ?faucet:faucet]
    Conditions at start: [Holding(?robot:robot, ?jug:jug), NoJugAtFaucet
    (?faucet:faucet)]
    Conditions overall: []
    Conditions at end: []
    Add Effects: [HandEmpty(?robot:robot), JugAtFaucet(?jug:jug, ?faucet:
    faucet)]
    Delete Effects: [Holding(?robot:robot, ?jug:jug), NoJugAtFaucet(?
    faucet:faucet)]
    Ignore Effects: []
    Log Strength: 1.0000
    Delay Distribution: DiscreteGaussianDelay(3.0000, 0.1000)
    Option Spec: PlaceUnderFaucet(?robot:robot, ?faucet:faucet),
```

```
EndogenousProcess-SwitchBurnerOff:
    Parameters: [?robot:robot, ?burner:burner]
    Conditions at start: [BurnerOn(?burner:burner), HandEmpty(?robot:
    robot)]
    Conditions overall: []
```

```
   Conditions at end: []
   Add Effects: [BurnerOff(?burner:burner)]
   Delete Effects: [BurnerOn(?burner:burner)]
   Ignore Effects: []
   Log Strength: 1.0000
   Delay Distribution: DiscreteGaussianDelay(1.0000, 0.1000)
   Option Spec: SwitchBurnerOff(?robot:robot, ?burner:burner),
```

```
EndogenousProcess-SwitchBurnerOn:
   Parameters: [?robot:robot, ?burner:burner]
   Conditions at start: [BurnerOff(?burner:burner), HandEmpty(?robot:
   robot)]
   Conditions overall: []
   Conditions at end: []
   Add Effects: [BurnerOn(?burner:burner)]
   Delete Effects: [BurnerOff(?burner:burner)]
   Ignore Effects: []
   Log Strength: 1.0000
   Delay Distribution: DiscreteGaussianDelay(3.0000, 0.1000)
   Option Spec: SwitchBurnerOn(?robot:robot, ?burner:burner),
```

```
EndogenousProcess-SwitchFaucetOff:
   Parameters: [?robot:robot, ?faucet:faucet]
   Conditions at start: [FaucetOn(?faucet:faucet), HandEmpty(?robot:
   robot)]
   Conditions overall: []
   Conditions at end: []
   Add Effects: [FaucetOff(?faucet:faucet)]
   Delete Effects: [FaucetOn(?faucet:faucet)]
   Ignore Effects: []
   Log Strength: 1.0000
   Delay Distribution: DiscreteGaussianDelay(1.0000, 0.1000)
   Option Spec: SwitchFaucetOff(?robot:robot, ?faucet:faucet),
```

```
EndogenousProcess-SwitchFaucetOn:
   Parameters: [?robot:robot, ?faucet:faucet]
   Conditions at start: [FaucetOff(?faucet:faucet), HandEmpty(?robot:
   robot)]
   Conditions overall: []
   Conditions at end: []
   Add Effects: [FaucetOn(?faucet:faucet)]
   Delete Effects: [FaucetOff(?faucet:faucet)]
   Ignore Effects: []
   Log Strength: 1.0000
   Delay Distribution: DiscreteGaussianDelay(1.0000, 0.1000)
   Option Spec: SwitchFaucetOn(?robot:robot, ?faucet:faucet)
```

**Additional predicates.** NoWaterSpilled, WaterBoiled, JugFilled, NoJugAtFaucetOrAtFaucetAndFilled

### B.4 DOMINO

**Train/Test split.** The training tasks takes place in a compact $3x2$ grid, where the agent must arrange one movable domino to successfully topple a single target domino. The test tasks are more complex in three ways: the workspace is enlarged to a $4x3$ grid, the number of movable dominoes is increased to two, and the goals may require toppling either one or two target dominoes.

**Goal predicates.** Toppled

**Initial predicates and endogenous processes.** Upright, InFrontDirection, InitialBlock, MovableBlock, Toppled, AdjacentTo, DominoAtPos, Holding, DominoAtRot, HandEmpty, Tilting, PosClear

```
EndogenousProcess-NoOp:
    Parameters: [?robot:robot]
    Conditions at start: []
    Conditions overall: []
    Conditions at end: []
    Add Effects: []
    Delete Effects: []
    Ignore Effects: [AdjacentTo, DominoAtPos, DominoAtRot, PosClear]
    Log Strength: 1.0000
    Delay Distribution: ConstantDelay(1.0000)
    Option Spec: NoOp(?robot:robot),
```

```
EndogenousProcess-PickDomino:
    Parameters: [?robot:robot, ?domino:domino, ?pos:loc, ?rot:angle]
    Conditions at start: [DominoAtPos(?domino:domino, ?pos:loc),
    DominoAtRot(?domino:domino, ?rot:angle), HandEmpty(?robot:robot),
    MovableBlock(?domino:domino), Upright(?domino:domino)]
    Conditions overall: []
    Conditions at end: []
    Add Effects: [Holding(?robot:robot, ?domino:domino), PosClear(?pos:
    loc)]
    Delete Effects: [DominoAtPos(?domino:domino, ?pos:loc), DominoAtRot(?
    domino:domino, ?rot:angle), HandEmpty(?robot:robot)]
    Ignore Effects: [DominoAtPos, DominoAtRot, PosClear, Tilting, Toppled
    , Upright]
    Log Strength: 1.0000
    Delay Distribution: DiscreteGaussianDelay(4.0000, 0.1000)
    Option Spec: Pick(?robot:robot, ?domino:domino),
```

```
EndogenousProcess-PlaceDomino:
    Parameters: [?robot:robot, ?domino1:domino, ?domino2:domino, ?pos1:
    loc, ?rot:angle]
    Conditions at start: [AdjacentTo(?pos1:loc, ?domino2:domino), Holding
    (?robot:robot, ?domino1:domino), PosClear(?pos1:loc), Upright(?
    domino2:domino)]
    Conditions overall: []
    Conditions at end: []
    Add Effects: [DominoAtPos(?domino1:domino, ?pos1:loc), DominoAtRot(?
    domino1:domino, ?rot:angle), HandEmpty(?robot:robot)]
    Delete Effects: [Holding(?robot:robot, ?domino1:domino), PosClear(?
    pos1:loc)]
    Ignore Effects: [AdjacentTo, DominoAtPos, DominoAtRot, PosClear,
    Tilting]
    Log Strength: 1.0000
    Delay Distribution: DiscreteGaussianDelay(3.0000, 0.1000)
    Option Spec: Place(?robot:robot, ?domino1:domino, ?domino2:domino, ?
    pos1:loc, ?rot:angle),
```

```
EndogenousProcess-PushStartBlock:
    Parameters: [?robot:robot, ?domino:domino]
    Conditions at start: [HandEmpty(?robot:robot), InitialBlock(?domino:
    domino), Upright(?domino:domino)]
    Conditions overall: []
    Conditions at end: []
    Add Effects: [Tilting(?domino:domino)]
    Delete Effects: [Upright(?domino:domino)]
    Ignore Effects: [AdjacentTo, DominoAtPos, DominoAtRot, PosClear]
    Log Strength: 1.0000
    Delay Distribution: DiscreteGaussianDelay(1.0000, 0.1000)
    Option Spec: Push(?robot:robot, ?domino:domino)
```

**Additional predicates.** {NotHeavy}

## B.5 FAN

**Train/Test split.** Training tasks are conducted on a small $3x3$ grid containing a single wall obstacle. In contrast, test tasks feature a larger $6x4$ grid and more intricate mazes constructed with either two or three walls.

**Goal predicates.** {BallAtLoc}

**Initial predicates and endogenous processes.** {SideOf, BallAtLoc, ClearLoc, FanOn, FanOff}

```
EndogenousProcess-NoOp:
    Parameters: [?robot:robot]
    Conditions at start: []
    Conditions overall: []
    Conditions at end: []
    Add Effects: []
    Delete Effects: []
    Ignore Effects: []
    Log Strength: 1.0000
    Delay Distribution: ConstantDelay(1.0000)
    Option Spec: NoOp(?robot:robot),
```

```
EndogenousProcess-TurnFanOff:
    Parameters: [?robot:robot, ?fan:fan]
    Conditions at start: [FanOn(?fan:fan)]
    Conditions overall: []
    Conditions at end: []
    Add Effects: [FanOff(?fan:fan)]
    Delete Effects: [FanOn(?fan:fan)]
    Ignore Effects: []
    Log Strength: 1.0000
    Delay Distribution: DiscreteGaussianDelay(2.0000, 0.1000)
    Option Spec: SwitchOff(?robot:robot, ?fan:fan),
```

```
EndogenousProcess-TurnFanOn:
    Parameters: [?robot:robot, ?fan:fan]
    Conditions at start: [FanOff(?fan:fan)]
    Conditions overall: []
    Conditions at end: []
    Add Effects: [FanOn(?fan:fan)]
    Delete Effects: [FanOff(?fan:fan)]
    Ignore Effects: []
    Log Strength: 1.0000
    Delay Distribution: DiscreteGaussianDelay(2.0000, 0.1000)
    Option Spec: SwitchOn(?robot:robot, ?fan:fan)
```

**Additional predicates.** {FanFacingSide, OppositeFan}

## C ADDITIONAL EXPERIMENT DETAILS

### C.1 LEARNED CAUSAL PROCESSES

We show example learned predicates and causal processes in each domain.

#### C.1.1 COFFEE

**Learned predicates and processes.** {JugFilled}

```
EndogenousProcess-NoOp:
    Parameters: [?robot:robot]
    Conditions at start: []
    Conditions overall: []
    Conditions at end: []
    Add Effects: []
    Delete Effects: []
    Ignore Effects: [JugAboveCup, NotAboveCup]
    Log Strength: -0.0113
    Delay Distribution: ConstantDelay(-0.0115)
    Option Spec: NoOp(?robot:robot)
```

```
EndogenousProcess-PickJugFromMachine:
    Parameters: [?robot:robot, ?jug:jug, ?machine:coffee_machine]
    Conditions at start: [HandEmpty(?robot:robot), JugInMachine(?jug:jug,
     ?machine:coffee_machine)]
    Conditions overall: []
    Conditions at end: []
    Add Effects: [Holding(?robot:robot, ?jug:jug)]
    Delete Effects: [HandEmpty(?robot:robot), JugInMachine(?jug:jug, ?
    machine:coffee_machine)]
    Ignore Effects: []
    Log Strength: 4.8335
    Delay Distribution: DiscreteGaussianDelay(13.8455, 5.3512)
    Option Spec: PickJug(?robot:robot, ?jug:jug)
```

```
EndogenousProcess-PickJugFromTable:
    Parameters: [?robot:robot, ?jug:jug]
    Conditions at start: [HandEmpty(?robot:robot), OnTable(?jug:jug)]
    Conditions overall: []
    Conditions at end: []
    Add Effects: [Holding(?robot:robot, ?jug:jug)]
    Delete Effects: [HandEmpty(?robot:robot), OnTable(?jug:jug)]
    Ignore Effects: []
    Log Strength: 1.5300
    Delay Distribution: DiscreteGaussianDelay(23.8392, 6.6450)
    Option Spec: PickJug(?robot:robot, ?jug:jug)
```

```
EndogenousProcess-PlaceJugInMachine:
    Parameters: [?robot:robot, ?jug:jug, ?machine:coffee_machine]
    Conditions at start: [Holding(?robot:robot, ?jug:jug), NotAboveCup(?
    robot:robot, ?jug:jug)]
    Conditions overall: []
    Conditions at end: []
    Add Effects: [HandEmpty(?robot:robot), JugInMachine(?jug:jug, ?
    machine:coffee_machine)]
    Delete Effects: [Holding(?robot:robot, ?jug:jug)]
    Ignore Effects: []
    Log Strength: 1.6979
    Delay Distribution: DiscreteGaussianDelay(20.0003, 6.5394)
    Option Spec: PlaceJugInMachine(?robot:robot, ?jug:jug, ?machine:
    coffee_machine)
```

```
EndogenousProcess-PourFromCup:
    Parameters: [?robot:robot, ?jug:jug, ?to_cup:cup, ?from_cup:cup]
    Conditions at start: [Holding(?robot:robot, ?jug:jug), JugAboveCup(?
    jug:jug, ?from_cup:cup)]
    Conditions overall: []
    Conditions at end: []
    Add Effects: [JugAboveCup(?jug:jug, ?to_cup:cup)]
    Delete Effects: [JugAboveCup(?jug:jug, ?from_cup:cup), NotAboveCup(?
    robot:robot, ?jug:jug)]
```

```
      Ignore Effects: [JugAboveCup, NotAboveCup]
      Log Strength: 0.0012
      Delay Distribution: DiscreteGaussianDelay(1.0125, 1.0112)
      Option Spec: Pour(?robot:robot, ?jug:jug, ?to_cup:cup)
```

```
EndogenousProcess-PourFromNotAboveCup:
    Parameters: [?robot:robot, ?jug:jug, ?cup:cup]
    Conditions at start: [Holding(?robot:robot, ?jug:jug), NotAboveCup(?
    robot:robot, ?jug:jug)]
    Conditions overall: []
    Conditions at end: []
    Add Effects: [JugAboveCup(?jug:jug, ?cup:cup)]
    Delete Effects: [NotAboveCup(?robot:robot, ?jug:jug)]
    Ignore Effects: []
    Log Strength: 1.7079
    Delay Distribution: DiscreteGaussianDelay(7.4837, 5.0596)
    Option Spec: Pour(?robot:robot, ?jug:jug, ?cup:cup)
```

```
EndogenousProcess-TurnMachineOn:
    Parameters: [?robot:robot, ?jug:jug, ?machine:coffee_machine]
    Conditions at start: [HandEmpty(?robot:robot), JugInMachine(?jug:jug,
     ?machine:coffee_machine)]
    Conditions overall: []
    Conditions at end: []
    Add Effects: [MachineOn(?machine:coffee_machine)]
    Delete Effects: []
    Ignore Effects: []
    Log Strength: 1.7298
    Delay Distribution: DiscreteGaussianDelay(18.3430, 6.4795)
    Option Spec: TurnMachineOn(?robot:robot, ?machine:coffee_machine)
```

```
ExogenousProcess-Op3:
    Parameters: [?x0:coffee_machine, ?x2:jug]
    Conditions at start: [JugInMachine(?x2:jug, ?x0:coffee_machine),
    MachineOn(?x0:coffee_machine)]
    Conditions overall: [JugInMachine(?x2:jug, ?x0:coffee_machine),
    MachineOn(?x0:coffee_machine)]
    Conditions at end: []
    Add Effects: [JugFilled(?x2:jug)]
    Delete Effects: []
    Log Strength: 1.7168
    Delay Distribution: DiscreteGaussianDelay(17.3098, 6.4991)
```

```
ExogenousProcess-Op5:
    Parameters: [?x1:cup, ?x2:jug, ?x3:robot]
    Conditions at start: [Holding(?x3:robot, ?x2:jug), JugAboveCup(?x2:
    jug, ?x1:cup), JugFilled(?x2:jug)]
    Conditions overall: [Holding(?x3:robot, ?x2:jug), JugAboveCup(?x2:jug
    , ?x1:cup), JugFilled(?x2:jug)]
    Conditions at end: []
    Add Effects: [CupFilled(?x1:cup)]
    Delete Effects: []
    Log Strength: 1.7250
    Delay Distribution: DiscreteGaussianDelay(4.5577, 1.8173)
```

### C.1.2  GROW

**Learned predicates and processes.**  {ColorMatches}

```
EndogenousProcess-NoOp:
    Parameters: [?robot:robot]
    Conditions at start: []
```

```
    Conditions overall: []
    Conditions at end: []
    Add Effects: []
    Delete Effects: []
    Ignore Effects: []
    Log Strength: -0.0113
    Delay Distribution: DiscreteGaussianDelay(25.6750, 6.9284)
    Option Spec: NoOp(?robot:robot)
```

```
EndogenousProcess-PickJugFromTable:
    Parameters: [?robot:robot, ?jug:jug]
    Conditions at start: [HandEmpty(?robot:robot), JugOnTable(?jug:jug)]
    Conditions overall: []
    Conditions at end: []
    Add Effects: [Holding(?robot:robot, ?jug:jug)]
    Delete Effects: [HandEmpty(?robot:robot), JugOnTable(?jug:jug)]
    Ignore Effects: []
    Log Strength: 2.3222
    Delay Distribution: DiscreteGaussianDelay(32.9836, 6.9427)
    Option Spec: PickJug(?robot:robot, ?jug:jug)
```

```
EndogenousProcess-PlaceJugOnTable:
    Parameters: [?robot:robot, ?jug:jug, ?cup:cup]
    Conditions at start: [Holding(?robot:robot, ?jug:jug), JugAboveCup(?
    jug:jug, ?cup:cup)]
    Conditions overall: []
    Conditions at end: []
    Add Effects: [HandEmpty(?robot:robot), JugOnTable(?jug:jug),
    NotAboveCup(?robot:robot, ?jug:jug)]
    Delete Effects: [Holding(?robot:robot, ?jug:jug), JugAboveCup(?jug:
    jug, ?cup:cup)]
    Ignore Effects: [HandEmpty, Holding, JugAboveCup, JugOnTable,
    NotAboveCup]
    Log Strength: 1.9316
    Delay Distribution: DiscreteGaussianDelay(24.9979, 6.6815)
    Option Spec: Place(?robot:robot, ?jug:jug)
```

```
EndogenousProcess-PourFromAboveCup:
    Parameters: [?robot:robot, ?jug:jug, ?from_cup:cup, ?to_cup:cup]
    Conditions at start: [Holding(?robot:robot, ?jug:jug), JugAboveCup(?
    jug:jug, ?from_cup:cup)]
    Conditions overall: []
    Conditions at end: []
    Add Effects: [JugAboveCup(?jug:jug, ?to_cup:cup)]
    Delete Effects: [JugAboveCup(?jug:jug, ?from_cup:cup)]
    Ignore Effects: [JugAboveCup, NotAboveCup]
    Log Strength: -0.0126
    Delay Distribution: DiscreteGaussianDelay(1.0035, 1.0031)
    Option Spec: Pour(?robot:robot, ?jug:jug, ?to_cup:cup)
```

```
EndogenousProcess-PourFromNotAboveCup:
    Parameters: [?robot:robot, ?jug:jug, ?cup:cup]
    Conditions at start: [Holding(?robot:robot, ?jug:jug), NotAboveCup(?
    robot:robot, ?jug:jug)]
    Conditions overall: []
    Conditions at end: []
    Add Effects: [JugAboveCup(?jug:jug, ?cup:cup)]
    Delete Effects: [NotAboveCup(?robot:robot, ?jug:jug)]
    Ignore Effects: [JugAboveCup, NotAboveCup]
    Log Strength: 2.2282
    Delay Distribution: DiscreteGaussianDelay(28.9585, 6.9970)
    Option Spec: Pour(?robot:robot, ?jug:jug, ?cup:cup)
```

```
ExogenousProcess-Op0:
    Parameters: [?x1:cup, ?x3:jug, ?x4:robot]
    Conditions at start: [ColorMatches(?x3:jug, ?x1:cup), CupOnTable(?x1:
    cup), Holding(?x4:robot, ?x3:jug), JugAboveCup(?x3:jug, ?x1:cup)]
    Conditions overall: [ColorMatches(?x3:jug, ?x1:cup), CupOnTable(?x1:
    cup), Holding(?x4:robot, ?x3:jug), JugAboveCup(?x3:jug, ?x1:cup)]
    Conditions at end: []
    Add Effects: [Grown(?x1:cup)]
    Delete Effects: []
    Log Strength: 1.2238
    Delay Distribution: DiscreteGaussianDelay(30.6220, 6.7903)
```

### C.1.3   BOIL

**Learned predicates and processes.**   {JugIsHot, JugIsFull, NotSpilling}

```
EndogenousProcess-NoOp:
    Parameters: [?robot:robot]
    Conditions at start: []
    Conditions overall: []
    Conditions at end: []
    Add Effects: []
    Delete Effects: []
    Ignore Effects: []
    Log Strength: -0.0113
    Delay Distribution: ConstantDelay(-0.0115)
    Option Spec: NoOp(?robot:robot)
```

```
EndogenousProcess-PickJugFromBurner:
    Parameters: [?robot:robot, ?jug:jug, ?burner:burner]
    Conditions at start: [HandEmpty(?robot:robot), JugAtBurner(?jug:jug,
    ?burner:burner)]
    Conditions overall: []
    Conditions at end: []
    Add Effects: [Holding(?robot:robot, ?jug:jug), NoJugAtBurner(?burner:
    burner)]
    Delete Effects: [HandEmpty(?robot:robot), JugAtBurner(?jug:jug, ?
    burner:burner)]
    Ignore Effects: []
    Log Strength: -0.0043
    Delay Distribution: DiscreteGaussianDelay(1.0085, 1.0069)
    Option Spec: PickJug(?robot:robot, ?jug:jug)
```

```
EndogenousProcess-PickJugFromFaucet:
    Parameters: [?robot:robot, ?jug:jug, ?faucet:faucet]
    Conditions at start: [HandEmpty(?robot:robot), JugAtFaucet(?jug:jug,
    ?faucet:faucet)]
    Conditions overall: []
    Conditions at end: []
    Add Effects: [Holding(?robot:robot, ?jug:jug), NoJugAtFaucet(?faucet:
    faucet)]
    Delete Effects: [HandEmpty(?robot:robot), JugAtFaucet(?jug:jug, ?
    faucet:faucet)]
    Ignore Effects: []
    Log Strength: 1.9823
    Delay Distribution: DiscreteGaussianDelay(23.5668, 6.6476)
    Option Spec: PickJug(?robot:robot, ?jug:jug)
```

```
EndogenousProcess-PickJugFromOutsideFaucetAndBurner:
    Parameters: [?robot:robot, ?jug:jug]
    Conditions at start: [HandEmpty(?robot:robot), JugNotAtBurnerOrFaucet
    (?jug:jug)]
```

```
    Conditions overall: []
    Conditions at end: []
    Add Effects: [Holding(?robot:robot, ?jug:jug)]
    Delete Effects: [HandEmpty(?robot:robot), JugNotAtBurnerOrFaucet(?jug
    :jug)]
    Ignore Effects: []
    Log Strength: 1.1899
    Delay Distribution: DiscreteGaussianDelay(43.4278, 6.8760)
    Option Spec: PickJug(?robot:robot, ?jug:jug)
```

```
EndogenousProcess-PlaceOnBurner:
    Parameters: [?robot:robot, ?jug:jug, ?burner:burner]
    Conditions at start: [Holding(?robot:robot, ?jug:jug), NoJugAtBurner
    (?burner:burner)]
    Conditions overall: []
    Conditions at end: []
    Add Effects: [HandEmpty(?robot:robot), JugAtBurner(?jug:jug, ?burner:
    burner)]
    Delete Effects: [Holding(?robot:robot, ?jug:jug), NoJugAtBurner(?
    burner:burner)]
    Ignore Effects: []
    Log Strength: 2.1507
    Delay Distribution: DiscreteGaussianDelay(21.9568, 6.6072)
    Option Spec: PlaceOnBurner(?robot:robot, ?burner:burner)
```

```
EndogenousProcess-PlaceOutsideFaucetAndBurner:
    Parameters: [?robot:robot, ?jug:jug]
    Conditions at start: [Holding(?robot:robot, ?jug:jug)]
    Conditions overall: []
    Conditions at end: []
    Add Effects: [HandEmpty(?robot:robot), JugNotAtBurnerOrFaucet(?jug:
    jug)]
    Delete Effects: [Holding(?robot:robot, ?jug:jug)]
    Ignore Effects: []
    Log Strength: -0.0025
    Delay Distribution: DiscreteGaussianDelay(0.9866, 0.9832)
    Option Spec: PlaceOutsideBurnerAndFaucet(?robot:robot)
```

```
EndogenousProcess-PlaceUnderFaucet:
    Parameters: [?robot:robot, ?jug:jug, ?faucet:faucet]
    Conditions at start: [Holding(?robot:robot, ?jug:jug), NoJugAtFaucet
    (?faucet:faucet)]
    Conditions overall: []
    Conditions at end: []
    Add Effects: [HandEmpty(?robot:robot), JugAtFaucet(?jug:jug, ?faucet:
    faucet)]
    Delete Effects: [Holding(?robot:robot, ?jug:jug), NoJugAtFaucet(?
    faucet:faucet)]
    Ignore Effects: []
    Log Strength: 1.9426
    Delay Distribution: DiscreteGaussianDelay(41.1660, 6.8798)
    Option Spec: PlaceUnderFaucet(?robot:robot, ?faucet:faucet)
```

```
EndogenousProcess-SwitchBurnerOff:
    Parameters: [?robot:robot, ?burner:burner]
    Conditions at start: [BurnerOn(?burner:burner), HandEmpty(?robot:
    robot)]
    Conditions overall: []
    Conditions at end: []
    Add Effects: [BurnerOff(?burner:burner)]
    Delete Effects: [BurnerOn(?burner:burner)]
    Ignore Effects: []
    Log Strength: 5.6200
```

```
      Delay Distribution: DiscreteGaussianDelay(9.8894, 4.0791)
      Option Spec: SwitchBurnerOff(?robot:robot, ?burner:burner)
```

```
EndogenousProcess-SwitchBurnerOn:
    Parameters: [?robot:robot, ?burner:burner]
    Conditions at start: [BurnerOff(?burner:burner), HandEmpty(?robot:
    robot)]
    Conditions overall: []
    Conditions at end: []
    Add Effects: [BurnerOn(?burner:burner)]
    Delete Effects: [BurnerOff(?burner:burner)]
    Ignore Effects: []
    Log Strength: 1.9554
    Delay Distribution: DiscreteGaussianDelay(32.0574, 6.7500)
    Option Spec: SwitchBurnerOn(?robot:robot, ?burner:burner)
```

```
EndogenousProcess-SwitchFaucetOff:
    Parameters: [?robot:robot, ?faucet:faucet]
    Conditions at start: [FaucetOn(?faucet:faucet), HandEmpty(?robot:
    robot)]
    Conditions overall: []
    Conditions at end: []
    Add Effects: [FaucetOff(?faucet:faucet)]
    Delete Effects: [FaucetOn(?faucet:faucet)]
    Ignore Effects: []
    Log Strength: 1.4501
    Delay Distribution: DiscreteGaussianDelay(27.5714, 6.7946)
    Option Spec: SwitchFaucetOff(?robot:robot, ?faucet:faucet)
```

```
EndogenousProcess-SwitchFaucetOn:
    Parameters: [?robot:robot, ?faucet:faucet]
    Conditions at start: [FaucetOff(?faucet:faucet), HandEmpty(?robot:
    robot)]
    Conditions overall: []
    Conditions at end: []
    Add Effects: [FaucetOn(?faucet:faucet)]
    Delete Effects: [FaucetOff(?faucet:faucet)]
    Ignore Effects: []
    Log Strength: 1.7156
    Delay Distribution: DiscreteGaussianDelay(35.2949, 6.7557)
    Option Spec: SwitchFaucetOn(?robot:robot, ?faucet:faucet)
```

```
ExogenousProcess-Op0:
    Parameters: [?x1:faucet, ?x2:jug]
    Conditions at start: [FaucetOn(?x1:faucet), JugAtFaucet(?x2:jug, ?x1:
    faucet)]
    Conditions overall: [FaucetOn(?x1:faucet), JugAtFaucet(?x2:jug, ?x1:
    faucet)]
    Conditions at end: []
    Add Effects: [JugIsFull(?x2:jug)]
    Delete Effects: []
    Log Strength: 1.2791
    Delay Distribution: DiscreteGaussianDelay(33.1148, 6.7873)
```

```
ExogenousProcess-Op1:
    Parameters: [?x0:burner, ?x2:jug]
    Conditions at start: [BurnerOn(?x0:burner), JugAtBurner(?x2:jug, ?x0:
    burner), JugIsFull(?x2:jug)]
    Conditions overall: [BurnerOn(?x0:burner), JugAtBurner(?x2:jug, ?x0:
    burner), JugIsFull(?x2:jug)]
    Conditions at end: []
    Add Effects: [JugIsHot(?x2:jug)]
```

```
    Delete Effects: []
    Log Strength: 1.9391
    Delay Distribution: DiscreteGaussianDelay(17.6401, 6.4452)
```

```
ExogenousProcess-Op2:
    Parameters: [?x0:burner, ?x1:faucet, ?x2:human, ?x3:jug, ?x4:robot]
    Conditions at start: [BurnerOff(?x0:burner), FaucetOff(?x1:faucet),
    HandEmpty(?x4:robot), JugIsFull(?x3:jug), JugIsHot(?x3:jug),
    NotSpilling(?x1:faucet)]
    Conditions overall: [BurnerOff(?x0:burner), FaucetOff(?x1:faucet),
    HandEmpty(?x4:robot), JugIsFull(?x3:jug), JugIsHot(?x3:jug),
    NotSpilling(?x1:faucet)]
    Conditions at end: []
    Add Effects: [HumanHappy(?x2:human, ?x3:jug, ?x0:burner)]
    Delete Effects: []
    Log Strength: 1.9474
    Delay Distribution: DiscreteGaussianDelay(5.5717, 4.1486)
```

```
ExogenousProcess-Op3:
    Parameters: [?x1:faucet]
    Conditions at start: [FaucetOn(?x1:faucet), NoJugAtFaucet(?x1:faucet)
    , NotSpilling(?x1:faucet)]
    Conditions overall: [FaucetOn(?x1:faucet), NoJugAtFaucet(?x1:faucet),
     NotSpilling(?x1:faucet)]
    Conditions at end: []
    Add Effects: []
    Delete Effects: [NotSpilling(?x1:faucet)]
    Log Strength: -0.0029
    Delay Distribution: DiscreteGaussianDelay(1.0005, 1.0052)
```

### C.1.4 DOMINO

**Learned predicates and processes.** {NOT-IsImmovable}

```
EndogenousProcess-NoOp:
    Parameters: [?robot:robot]
    Conditions at start: []
    Conditions overall: []
    Conditions at end: []
    Add Effects: []
    Delete Effects: []
    Ignore Effects: [AdjacentTo, DominoAtPos, DominoAtRot, PosClear]
    Log Strength: 1.0000
    Delay Distribution: ConstantDelay(1.0000)
    Option Spec: NoOp(?robot:robot)
```

```
EndogenousProcess-PickDomino:
    Parameters: [?robot:robot, ?domino:domino, ?pos:loc, ?rot:angle]
    Conditions at start: [DominoAtPos(?domino:domino, ?pos:loc),
    DominoAtRot(?domino:domino, ?rot:angle), HandEmpty(?robot:robot),
    MovableBlock(?domino:domino), Upright(?domino:domino)]
    Conditions overall: []
    Conditions at end: []
    Add Effects: [Holding(?robot:robot, ?domino:domino), PosClear(?pos:
    loc)]
    Delete Effects: [DominoAtPos(?domino:domino, ?pos:loc), DominoAtRot(?
    domino:domino, ?rot:angle), HandEmpty(?robot:robot)]
    Ignore Effects: [DominoAtPos, DominoAtRot, PosClear, Tilting, Toppled
    , Upright]
    Log Strength: 0.0000
    Delay Distribution: DiscreteGaussianDelay(14.0000, 0.1000)
    Option Spec: Pick(?robot:robot, ?domino:domino)
```

```
EndogenousProcess-PlaceDomino:
   Parameters: [?robot:robot, ?domino1:domino, ?domino2:domino, ?pos1:
   loc, ?rot:angle]
   Conditions at start: [AdjacentTo(?pos1:loc, ?domino2:domino), Holding
   (?robot:robot, ?domino1:domino), PosClear(?pos1:loc), Upright(?
   domino2:domino)]
   Conditions overall: []
   Conditions at end: []
   Add Effects: [DominoAtPos(?domino1:domino, ?pos1:loc), DominoAtRot(?
   domino1:domino, ?rot:angle), HandEmpty(?robot:robot)]
   Delete Effects: [Holding(?robot:robot, ?domino1:domino), PosClear(?
   pos1:loc)]
   Ignore Effects: [AdjacentTo, DominoAtPos, DominoAtRot, PosClear,
   Tilting]
   Log Strength: 0.0000
   Delay Distribution: DiscreteGaussianDelay(8.0000, 0.1000)
   Option Spec: Place(?robot:robot, ?domino1:domino, ?domino2:domino, ?
   pos1:loc, ?rot:angle)
```

```
EndogenousProcess-PushStartBlock:
   Parameters: [?robot:robot, ?domino:domino]
   Conditions at start: [HandEmpty(?robot:robot), InitialBlock(?domino:
   domino), Upright(?domino:domino)]
   Conditions overall: []
   Conditions at end: []
   Add Effects: [Tilting(?domino:domino)]
   Delete Effects: [Upright(?domino:domino)]
   Ignore Effects: [AdjacentTo, DominoAtPos, DominoAtRot, PosClear]
   Log Strength: 0.0000
   Delay Distribution: DiscreteGaussianDelay(8.0000, 0.1000)
   Option Spec: Push(?robot:robot, ?domino:domino)
```

```
ExogenousProcess-Op0:
   Parameters: [?x1:domino, ?x2:domino, ?x11:direction]
   Conditions at start: [InFrontDirection(?x1:domino, ?x2:domino, ?x11:
   direction), NOT-IsImmovable(?x1:domino), NOT-IsImmovable(?x2:domino),
    Tilting(?x1:domino), Upright(?x2:domino)]
   Conditions overall: [InFrontDirection(?x1:domino, ?x2:domino, ?x11:
   direction), NOT-IsImmovable(?x1:domino), NOT-IsImmovable(?x2:domino),
    Tilting(?x1:domino), Upright(?x2:domino)]
   Conditions at end: []
   Add Effects: [Tilting(?x2:domino)]
   Delete Effects: []
   Log Strength: 0.0000
   Delay Distribution: DiscreteGaussianDelay(2.0000, 0.1000)
```

```
ExogenousProcess-Op1:
   Parameters: [?x1:domino, ?x2:domino, ?x11:direction]
   Conditions at start: [InFrontDirection(?x1:domino, ?x2:domino, ?x11:
   direction), NOT-IsImmovable(?x1:domino), NOT-IsImmovable(?x2:domino),
    Tilting(?x1:domino), Upright(?x2:domino)]
   Conditions overall: [InFrontDirection(?x1:domino, ?x2:domino, ?x11:
   direction), NOT-IsImmovable(?x1:domino), NOT-IsImmovable(?x2:domino),
    Tilting(?x1:domino), Upright(?x2:domino)]
   Conditions at end: []
   Add Effects: []
   Delete Effects: [Upright(?x2:domino)]
   Log Strength: 0.0000
   Delay Distribution: DiscreteGaussianDelay(2.0000, 0.1000)
```

```
ExogenousProcess-Op2:
   Parameters: [?x1:domino]
```

```
        Conditions at start: [Tilting(?x1:domino)]
        Conditions overall: [Tilting(?x1:domino)]
        Conditions at end: []
        Add Effects: []
        Delete Effects: [Tilting(?x1:domino)]
        Log Strength: 0.0000
        Delay Distribution: DiscreteGaussianDelay(5.0000, 0.1000)
```

```
ExogenousProcess-Op3:
        Parameters: [?x1:domino]
        Conditions at start: [Tilting(?x1:domino)]
        Conditions overall: [Tilting(?x1:domino)]
        Conditions at end: []
        Add Effects: [Toppled(?x1:domino)]
        Delete Effects: []
        Log Strength: 0.0000
        Delay Distribution: DiscreteGaussianDelay(5.0000, 0.1000)
```

### C.1.5 FAN

**Learned predicates and processes.** {FanFaces}

```
EndogenousProcess-NoOp:
        Parameters: [?robot:robot]
        Conditions at start: []
        Conditions overall: []
        Conditions at end: []
        Add Effects: []
        Delete Effects: []
        Ignore Effects: []
        Log Strength: 0.0000
        Delay Distribution: ConstantDelay(0.0000)
        Option Spec: NoOp(?robot:robot)
```

```
EndogenousProcess-TurnFanOff:
        Parameters: [?robot:robot, ?fan:fan]
        Conditions at start: [FanOn(?fan:fan)]
        Conditions overall: []
        Conditions at end: []
        Add Effects: [FanOff(?fan:fan)]
        Delete Effects: [FanOn(?fan:fan)]
        Ignore Effects: []
        Log Strength: 0.0000
        Delay Distribution: DiscreteGaussianDelay(11.0000, 0.1000)
        Option Spec: SwitchOff(?robot:robot, ?fan:fan)
```

```
EndogenousProcess-TurnFanOn:
        Parameters: [?robot:robot, ?fan:fan]
        Conditions at start: [FanOff(?fan:fan)]
        Conditions overall: []
        Conditions at end: []
        Add Effects: [FanOn(?fan:fan)]
        Delete Effects: [FanOff(?fan:fan)]
        Ignore Effects: []
        Log Strength: 0.0000
        Delay Distribution: DiscreteGaussianDelay(13.6667, 2.0817)
        Option Spec: SwitchOn(?robot:robot, ?fan:fan)
```

```
ExogenousProcess-Op1:
        Parameters: [?x0:ball, ?x2:fan, ?x3:fan, ?x14:loc, ?x15:loc, ?x16:
        side]
```

```
Conditions at start: [BallAtLoc(?x0:ball, ?x14:loc), ClearLoc(?x15:
loc), FanFaces(?x2:fan, ?x16:side), FanOff(?x3:fan), FanOn(?x2:fan),
SideOf(?x15:loc, ?x14:loc, ?x16:side)]
Conditions overall: [BallAtLoc(?x0:ball, ?x14:loc), ClearLoc(?x15:loc
), FanFaces(?x2:fan, ?x16:side), FanOff(?x3:fan), FanOn(?x2:fan),
SideOf(?x15:loc, ?x14:loc, ?x16:side)]
Conditions at end: []
Add Effects: [BallAtLoc(?x0:ball, ?x15:loc)]
Delete Effects: [BallAtLoc(?x0:ball, ?x14:loc)]
Log Strength: 0.0000
Delay Distribution: DiscreteGaussianDelay(21.5000, 8.2407)
```

## C.2 FURTHER LEARNING AND PLANNING STATISTICS

Each online learning iteration of ExoPredicator takes approximately 30–360 seconds wall-clock time depending on the domain, using 32 CPU cores and no GPU. In comparison, the HRL baseline requires on average about 600 seconds per online iteration, and its runtime increases as it accumulates more data (taking longer to converge), reaching up to 3200 seconds in some cases. The VLM planning baseline does not perform online learning, but can incur substantial test-time cost because it issues computationally expensive VLM calls every time it generates a plan.

Monetaryly, assuming Gemini 2.5 Pro with standard pricing, a representative online iteration consuming a combined 10,000 input tokens and 1,000 output tokens would cost approximately \$0.0225 in total: \$0.0125 for inputs (10,000 / 1,000,000 × 1.25) and \$0.01 for outputs (1,000 / 1,000,000 × 10).

The following table lists the success rate and planning time statistics.

| Environment | Manual | | Ours | | No invent | |
|---|---|---|---|---|---|---|
| | Succ | Time | Succ | Time | Succ | Time |
| Coffee | 99.3 | 0.612 | 99.3 | 0.851 | 0.0 | – |
| Grow | 92.0 | 0.608 | 93.3 | 0.922 | 0.0 | – |
| Boil | 100.0 | 15.467 | 92.7 | 12.204 | 0.0 | – |
| Domino | 97.3 | 31.710 | 98.7 | 21.299 | 62.0 | 0.000 |
| Fan | 97.3 | 16.143 | 97.3 | 58.244 | 0.0 | – |

## C.3 PRIORS AND DISTRIBUTIONS

In all experiments, we model process delays with a truncated discrete Gaussian distribution over positive integers $\{1, \ldots, 300\}$. The distribution is parameterized by a log-mean and a log-standard-deviation.

The log-mean, log-standard-deviation, and log-process-weights are initialized from a normal distribution with mean $0$ and standard deviation $0.01$.

