# OpenReview forum: "ExoPredicator: Learning Abstract Models of Dynamic Worlds for Robot Planning"
_ICLR.cc/2026/Conference — ICLR 2026 Poster_

### Official Review · Reviewer_Fikj · 2025-11-01

**Soundness:** 3
**Presentation:** 3
**Contribution:** 3
**Rating:** 8
**Confidence:** 3

**Summary:**

This paper introduces ExoPredicator, a framework for learning abstract world models that capture both endogenous (agent-controlled) and exogenous (environment-driven) causal processes. Unlike standard world models or symbolic planners that assume all changes stem from the agent’s actions, ExoPredicator explicitly models external processes that unfold concurrently over time (e.g., water boiling after switching on a kettle).

The framework represents states as collections of predicates grounded by vision-language models (VLMs) and describes temporal dynamics via causal processes—structured tuples encoding conditions, effects, and probabilistic delay distributions. The system jointly learns (i) state abstractions (via LLM-guided predicate invention) and (ii) causal process models (via Bayesian structure and parameter learning with variational inference).

Experiments across five simulated tabletop robotics environments (Coffee, Grow, Boil, Domino, and Fan) show that ExoPredicator learns accurate symbolic models from limited data, enabling efficient A* planning over “big-step” transitions and outperforming hierarchical RL, VLM-planning, and STRIPS-style baselines. Ablations indicate that both Bayesian model selection and LLM guidance are crucial for performance.

**Strengths:**

1. Strong motivation and relevance. The paper tackles the important problem of integrating state and temporal abstraction in world modeling, particularly addressing exogenous processes that evolve independently of agent actions.
2. Elegant conceptual framing. The formulation of causal processes as structured tuples with probabilistic delays provides a clean and general formalism for modeling both endogenous and exogenous dynamics.
3. Expressive yet practical abstraction. Defining abstract states through predicates queried via VLMs strikes a balance between symbolic expressivity and perceptual grounding.
4. Clear mechanistic interpretation. The notion of a big-step transition function is intuitive and well-motivated, allowing the planner to reason at a temporally coarse level.
5. The use of a variational lower bound for parameter estimation and Bayesian model selection for structure learning supports data-efficient model discovery.
6. Innovative use of LLMs. The integration of LLMs for proposing new symbolic forms of causal processes (Sec. 5.2) and predicates (Sec. 5.3) is a creative and well-executed design choice.
7. Well-designed experiments. The set of five environments captures a diverse range of exogenous phenomena, and the ablation studies provide useful insight into which components matter most.
8. Empirical results are convincing. ExoPredicator achieves strong generalization to unseen tasks, often outperforming baselines despite limited training data.

**Weaknesses:**

1. Questionable novelty claim. The paper argues that prior models (e.g., Dreamer, world models) “do not consider exogenous processes.” This is somewhat overstated. Model-based RL frameworks routinely model all sources of state transition—including environment-driven stochasticity.
2.  Overstatement regarding options/skills. The claim that options or skills cannot capture the temporal evolution of exogenous dynamics may be too strong. In principle, semi-MDP formulations with temporally extended actions can incorporate control-exogenous effects into the transition model, although this is not their main focus.
3. Assumption of known skill set. The framework assumes the availability of predefined high-level skills (Pick, Place, etc.), which limits autonomy and may constrain generalization to new embodiments.
4. Limited empirical diversity. While the environments are well-constructed, they are all tabletop simulation tasks with relatively low-dimensional perceptual spaces. It remains unclear how well the method scales to real-world or visually richer domains.

**Questions:**

1. What is the practical motivation of the frame axiom potential for enforcing state persistence over time?
2. Variational Representation: How are the variational distributions in sec. 5.1 represented and parameterized in practice?
3. When prompting the LLM to propose symbolic forms for exogenous processes, what kinds of templates or grammatical constraints are imposed?
4. Minimum Description Length: How is the MDL prior or description length of a causal process computed? Is it based on symbolic length, number of predicates, or something else?

---

> ### Author Response · Authors · 2025-11-19
>
> We thank the reviewer for the very positive assessment and thoughtful suggestions.
>
> > Questionable novelty claim. The paper argues that prior models (e.g., Dreamer, world models) “do not consider exogenous processes.” This is somewhat overstated. Model-based RL frameworks routinely model all sources of state transition—including environment-driven stochasticity.
> >
>
> > Overstatement regarding options/skills. The claim that options or skills cannot capture the temporal evolution of exogenous dynamics may be too strong. In principle, semi-MDP formulations with temporally extended actions can incorporate control-exogenous effects into the transition model, although this is not their main focus.
> >
>
> We agree our original wording was too strong. Our goal is **not** to claim that Dreamer-style world models or options *ignore* environment-driven stochasticity, but that they typically do **not explicitly represent exogenous processes as separate, temporally extended mechanisms** with their own activation conditions and delay distributions.
>
> We have revised the related work to clarify this:
>
> - We now state that prior abstract world models “**typically abstract dynamics through agent-centric actions, without explicitly modeling exogenous causal processes that unfold over time in the background**,” rather than claiming they “do not consider” exogenous processes.
> - In the HRL paragraph, we softened the language around options/skills and now emphasize that our approach is **complementary**: we keep the (semi-)MDP option view, but add an explicit layer of **named causal processes with stochastic delays** that can continue to evolve while options are executed. This makes the distinction between *folding* exogenous dynamics into the one-step transition vs. *explicitly modeling* them as concurrent processes more precise.
>
> > Assumption of known skill set. The framework assumes the availability of predefined high-level skills (Pick, Place, etc.), which limits autonomy and may constrain generalization to new embodiments.
> >
>
> We agree this is a limitation, and we now highlight it in the conclusion. Conceptually, we view a fixed skill library as a natural abstraction boundary: both humans and many robotic systems enter new environments with pre-trained motor primitives (Pick/Place/Push, SwitchOn/Off, …), yet still need to learn how the **world** behaves under those skills. Our contribution is to show that, even under this modest assumption, jointly learning **state and temporal abstractions** with exogenous processes yields strong generalization.
>
> We explicitly note that **combining ExoPredicator with skill discovery or option-learning methods** is a promising direction for future work, rather than claiming to solve end-to-end skill + world model learning.
>
> > Limited empirical diversity. While the environments are well-constructed, they are all tabletop simulation tasks with relatively low-dimensional perceptual spaces. It remains unclear how well the method scales to real-world or visually richer domains.
> >
>
> Our primary aim in this paper is a controlled first study of abstract world models that jointly handle **state abstraction, temporal abstraction, and exogenous processes**. The five environments (Coffee, Grow, Boil, Domino, Fan) were deliberately designed to exercise **different kinds of exogenous dynamics** (heating, growth, spills, cascades, wind-driven motion) while keeping low-level control manageable, so that we can attribute gains to the proposed model rather than to domain-specific engineering.
>
> We agree that scaling to real robots and richer perception is an important next step. We have expanded Sec. 6 and the conclusion to explicitly discuss these deployment challenges and to position our current results as a foundation for such follow-up work.
>
> ---
>
> > What is the practical motivation of the frame axiom potential for enforcing state persistence over time?
> >
>
> The frame potential is our way of encoding an **inertia prior**: most abstract facts should persist unless there is clear evidence that a causal process has changed them. Practically, this has two benefits:
>
> (1) It regularizes learning by discouraging explanations in which many predicates flip arbitrarily between timesteps; instead, changes are attributed to a small number of causal processes with well-defined conditions and delays. This makes it much harder for the model to “explain” data via spurious processes that constantly toggle predicates.
>
> (2) It yields a coherent probabilistic semantics over full trajectories, which in principle supports richer inference queries beyond planning. For example (though we do not explore this in the paper), given two support trajectories $x_1, x_2$ from slightly different environments and a test trajectory $y$, one can ask which environment $y$ is more likely to have come from under the learned process models. The persistence prior makes such comparisons well-posed by discouraging gratuitous predicate changes.

---

> > ### Author Response · Authors · 2025-11-19
> >
> > > Variational Representation: How are the variational distributions in sec. 5.1 represented and parameterized in practice?
> > >
> >
> > For variational inference, we use a **categorical** variational distribution over the discrete support of arrival times. Concretely, for each ground process and potential cause time, we maintain logits over possible arrival times and optimize these logits directly to maximize the ELBO. We have clarified this in Apps. A.3 and C.3.
> >
> > > When prompting the LLM to propose symbolic forms for exogenous processes, what kinds of templates or grammatical constraints are imposed?
> > >
> >
> > The full prompt templates are listed in Apps. A.3–A.4. We do not impose strong grammatical constraints beyond a simple structured format (e.g., returning condition sets as index lists over candidate atoms). After generation, we discard any malformed outputs or ones that cannot be executed on all states. In other words, we keep a proposed predicate only if the returned code can be run successfully on all states; otherwise it is filtered out.
> >
> > > Minimum Description Length: How is the MDL prior or description length of a causal process computed? Is it based on symbolic length, number of predicates, or something else?
> > >
> >
> > The MDL-style prior is based on the **number of atoms in the condition**. Specifically, we penalize each candidate process by a term proportional to $|C|$, the size of its condition set, with weight $10^{-4}$. This is the same predicate-size penalty used in prior abstraction-learning work (e.g., Liang et al., 2024; Silver et al., 2023).

---

### Official Review · Reviewer_Epsw · 2025-11-01

**Soundness:** 3
**Presentation:** 3
**Contribution:** 3
**Rating:** 8
**Confidence:** 3

**Summary:**

The paper introduces ExoPredicator, a framework for learning abstract world models that capture both endogenous skills and exogenous causal processes with stochastic and possibly delayed effects. The model jointly learns a symbolic state abstraction via predicate invention and a library of causal processes whose conditions, effects, and delay distributions are inferred from limited trajectory data using variational Bayesian inference guided by language model proposals. Planning operates in big steps with an A star search over high level actions, including a NoOp action that allows the agent to wait for exogenous dynamics to unfold. Across five PyBullet tabletop domains Coffee, Grow, Boil, Domino, and Fan the method achieves high solve rates and generalizes to harder held out tasks with more objects and more complex goals. It outperforms vision language planning, hierarchical RL, and STRIPS style operator learning baselines and includes ablations that highlight the roles of Bayesian model selection and LLM guidance.

**Strengths:**

- Novel problem framing that explicitly models exogenous processes with stochastic and delayed effects within an abstract world model. This goes beyond standard STRIPS or option based abstractions that assume instantaneous effects and agent centric dynamics.
- Integrated learning and planning system. The method couples predicate invention, Bayesian structure and parameter learning for causal processes, and a big step planner that reasons over concurrent exogenous dynamics via a NoOp wait action.
- Principled inference. The paper derives a variational objective using arrival time latent variables to avoid fine grained simulation and learns both process weights and delay distributions. The planning semantics are precise and the big step transition is specified with care.
- Empirical breadth and generalization. Strong results across five simulated domains with generalization to harder held out tasks. Convergence in at most three online iterations with only one or two demonstration seeded training tasks per seed shows good sample efficiency.
- Comparisons and ablations. Consistent improvements over ViLA, MAPLE, and VisPred. Ablations show that both LLM guidance and Bayesian model selection matter. Manual delay parameters underperform learned ones which supports the value of the inference component.
- Clarity of problem setup. The paper carefully distinguishes endogenous skills from exogenous processes and motivates why temporal abstraction for world dynamics is needed for long horizon planning.

**Weaknesses:**

- Evaluation is limited to simulated tabletop worlds with built in closed loop skills. There is no validation on real robots or with noisy perception which leaves open questions about robustness and deployment.
- Only three random seeds and a modest number of training tasks are used. While this highlights sample efficiency it makes statistical confidence harder to assess. Confidence intervals and significance tests are not reported.
- The approach relies on a fixed library of low level skills and on a proprietary LLM for proposals. This raises reproducibility concerns and limits analysis of end to end learning without pre specified skills.
- The failure case in Boil indicates that learned conditions can miss important disjunctive structure. More stress tests on overlapping or interacting exogenous processes would strengthen the case.

**Questions:**

- Clarify the assumed family for delay distributions. Are they discrete on timesteps or continuous then discretized. What priors are used for these distributions and for process weights.
- Provide more details on the segmentation and clustering used before process discovery. How sensitive is learning to the clip length and the clustering hyperparameters.
- Report wall clock times per online iteration and per plan for each environment. Include counts of LLM calls and average cost to understand practicality.
- How does the planner choose the duration to wait with NoOp. Is the wait until first abstract change always optimal or could waiting through an intermediate change be beneficial.
- How does the framework prevent spurious processes from being added when predicates are noisy. Is there any calibration of VLM predicate accuracy or robustness to false positives and false negatives.

---

> ### Author Response · Authors · 2025-11-19
>
> We thank the reviewer for the encouraging and detailed feedback, and for highlighting both the conceptual framing and the empirical strengths of our work.
>
> > Evaluation is limited to simulated tabletop worlds with built in closed loop skills. There is no validation on real robots or with noisy perception which leaves open questions about robustness and deployment.
> >
>
> Our goal in this work is to take a first step toward learning abstract world models that jointly handle state and temporal abstraction in the presence of exogenous processes. We chose simulated tabletop domains because they allow us to systematically vary objects, goals, and exogenous dynamics (e.g., Boil vs. Domino vs. Fan) while keeping low-level control simple, making it possible to isolate the contribution of the proposed model. We agree that deploying ExoPredicator on real robots and in settings with richer visual noise is an important next step, and we have added and highlighted a discussion of these deployment challenges and opportunities to the conclusion.
>
> > Only three random seeds and a modest number of training tasks are used. While this highlights sample efficiency it makes statistical confidence harder to assess. Confidence intervals and significance tests are not reported.
> >
>
> We agree that statistical reporting is important. Our plots in Fig. 6 already report confidence intervals across the three random seeds (we now state this explicitly in the caption).  As the reviewer notes, our design intentionally uses *few* training tasks and demonstrations to highlight sample efficiency and generalization to harder held-out tasks (more objects and more complex goals). We have already started running experiments with additional random seeds and will update the manuscript and plotting script as soon as these runs complete; preliminary results are consistent with the current curves. We view increasing the number of seeds and training tasks as complementary to our focus, and the existing error bars already indicate that performance is stable across seeds.
>
> > The approach relies on a fixed library of low level skills and on a proprietary LLM for proposals. This raises reproducibility concerns and limits analysis of end to end learning without pre specified skills.
> >
>
> We agree that the assumption of a fixed skill library is a limitation, but we view it as a natural abstraction boundary: both humans and many robotic systems enter new environments with pre-trained motor primitives (e.g., Pick/Place/Push) and must still learn how the *world* behaves under those skills. Learning an abstract world model with exogenous processes on top of such skills is, we believe, already a non-trivial and practically relevant problem. That said, combining ExoPredicator with skill discovery or end-to-end option learning is an exciting direction, and we have added this to the future-work discussion.
> Regarding reproducibility: (i) the framework is *model-agnostic* with respect to the LLM/VLM used—the only requirement is the ability to propose predicates and candidate condition sets; (ii) we will release code, including the full skill library, prompts, and process/predicate APIs, to make it easy to reproduce our experiments with different foundation models, including open-weight alternatives.
>
> > The failure case in Boil indicates that learned conditions can miss important disjunctive structure. More stress tests on overlapping or interacting exogenous processes would strengthen the case.
> >
>
> We agree that the Boil failure case exposes an important limitation of our current learning setup. We now make this limitation explicit in Sec. 6. The Boil domain was designed precisely to probe such complex interactions. Extending our process representation and learning procedure to better handle disjunctive preconditions, and testing it on more complex environments with overlapping exogenous processes, is a natural next step; we see the Boil failure as a constructive signal about where the model and predicate classes should be enriched.
>
> > Clarify the assumed family for delay distributions. Are they discrete on timesteps or continuous then discretized. What priors are used for these distributions and for process weights.
> >
>
> Delays are modeled as *discrete* Gaussian distributions over positive integer timesteps (support truncated at 300). The distribution is parameterized by log-mean and log-standard-deviation, and we place Gaussian priors (mean 0, std 0.01) on the log-mean, log-std, and process strength weights $W_L$. For variational inference, we use a categorical variational distribution over the discrete delay support (initialized to uniform). We now also clarify this in  Sec. 5.1, App. A.3., and App. C.3.

---

> ### Author Response · Authors · 2025-11-19
>
> > Provide more details on the segmentation and clustering used before process discovery. How sensitive is learning to the clip length and the clustering hyperparameters.
> >
>
> We have expanded App. A.4 to clarify this procedure. In our implementation, **clip length (the number of low-level steps in each trajectory clip) is not a tunable hyperparameter**. Each low-level trajectory is segmented whenever the abstract state changes: a segment starts at one abstract state and ends at the next time that the set of true predicates changes.
>
> The **clustering step also has no free hyperparameters**. Each segment is mapped to a discrete “change signature” describing which abstract predicates changed between its start and end states. Segments with the same signature are grouped together, and process learning is performed separately for each signature. This is equivalent to clustering with a deterministic, discrete key, so there is no dependence on distance thresholds, number of clusters, or similar hyperparameters. We now explicitly state this in App. A.4.
>
> > Report wall clock times per online iteration and per plan for each environment. Include counts of LLM calls and average cost to understand practicality.
> >
>
> We have added these details to App. C.2. Across domains, a single online learning iteration of **ExoPredicator** takes roughly 30–360 seconds of wall-clock time on 32 CPU cores (no GPU). By comparison, the HRL baseline (MAPLE) requires on average ~600 seconds per online iteration on a GPU, and its runtime grows substantially as it gathers more data, reaching up to ~3200 seconds in some runs. The VLM planning baseline (ViLA) does not perform online learning, but can incur significant *test-time* cost because it issues expensive VLM calls whenever it generates a plan.
>
> We also report per-episode planning times in App. C.2: once trained, ExoPredicator’s planning time is on the same order as the Manual oracle (e.g., about a second in Coffee/Grow and tens of seconds in Domino/Fan). Because ExoPredicator converges in at most three online iterations in all domains, the overall computational cost remains moderate.
>
> Regarding LLM usage and monetary cost: per online iteration, our method makes at most one VLM call for predicate invention and one LLM call for process-condition proposals, using roughly 10k input tokens and 1k output tokens in total. Under Gemini 2.5 Pro’s standard pricing, this corresponds to approximately **$0.023 per iteration**.
>
> > How does the planner choose the duration to wait with NoOp. Is the wait until first abstract change always optimal or could waiting through an intermediate change be beneficial.
> >
>
> This is a very insightful question. In the current implementation, the NoOp process simply waits until the **first** abstract state change, at which point $T_{\text{big}}$ returns control to the planner (Sec. 4, App. A.1). We agree that, in principle, it may be beneficial to wait through intermediate changes (e.g., to ride out transient or noisy effects); extending $T_{\text{big}}$ to reason about what to wait for is a promising direction.
>
> > How does the framework prevent spurious processes from being added when predicates are noisy. Is there any calibration of VLM predicate accuracy or robustness to false positives and false negatives.
> >
>
> We address spurious noisy predicates through predicate subset selection. Predicate invention proposes a larger pool of candidates, but only a subset is retained by optimizing a planning-based objective (Sec. 5.3,). Predicates that are too noisy or uninformative for planning will not improve this objective and are therefore dropped. This acts as an implicit robustness filter.

---

### Official Review · Reviewer_p9tB · 2025-11-01

**Soundness:** 3
**Presentation:** 1
**Contribution:** 2
**Rating:** 2
**Confidence:** 2

**Summary:**

This paper tackles long-horizon planning in dynamic environments where changes arise both from an agent’s actions and from independent external processes. It proposes a framework for learning abstract world models that capture symbolic state representations and causal processes with delayed effects.

**Strengths:**

The paper addresses a timely and important problem in robotics: learning abstract world models that can reason over both agent-driven and environment-driven dynamics. The motivation is clear and well-grounded, and the overall direction aligns with the community’s growing interest in long-horizon planning.  Experiments on simulated tabletop tasks show improved generalization compared to several baselines. However, the framework’s current complexity and heavy mathematical formulation limit its accessibility. A major revision to simplify the presentation or improve clarity would make the contribution more broadly useful.

**Weaknesses:**

- Overly complex framework and poor readability. The methodology is overloaded with notation and nested definitions that make it extremely difficult to follow. Many variables and terms are introduced without clear motivation, explanation, or citation. It is unclear which components are novel and which are borrowed or adapted from prior works. While the motivation in the abstract and introduction is sound, the formalism feels unnecessarily complicated for the problem being addressed.

- Insufficient experimental validation. The experiments are minimal and do not justify the complexity of the proposed framework. The evaluation focuses mainly on overall success rate, without ablations or detailed analyses that could demonstrate the contribution of each component. More fine-grained experiments are needed to show why the proposed system must be this complex and how each module contributes to its performance.

- Conceptual confusion around endogenous vs. exogenous processes. The paper defines exogenous processes as those that occur independently of the agent’s actions, yet uses “filling water” as an example, which is directly caused by the agent turning on the faucet. This blurs the distinction between endogenous and exogenous mechanisms. In practice, the framework seems to capture delayed consequences of the agent’s own actions rather than truly external processes. The paper would benefit from a clearer conceptual definition and concrete examples that distinguish these cases.

**Questions:**

See Weaknesses

---

> ### Author Response · Authors · 2025-11-19
>
> We thank the reviewer for the critical feedback. We appreciate that you found the problem motivation "clear and well-grounded" and the results strong. We have taken your concerns regarding readability and definitions very seriously and have significantly revised the manuscript to address them.
>
> > Overly complex framework and poor readability.
> >
>
> We appreciate the concern about readability. ExoPredicator does contain several components, but they all serve a single integrated learning and planning pipeline. The main components are:
>
> (1) learn a symbolic state abstraction with predicates,
>
> (2) learn a library of causal processes (endogenous and exogenous) with probabilistic delays, and
>
> (3) plan using a big-step transition model over these abstractions.
>
> To make this clearer, we have revised the paper to better separate high-level ideas from mathematical details and to connect the notation to concrete examples:
>
> - **Clearer high-level story in Sec. 3.** We now explicitly emphasize that an abstract state is “the collection of symbolic facts the agent believes about the world” and note that predicate-based state abstraction is standard in robot task planning, with citation (Garrett et al., 2021). This anchors our notation in familiar TAMP-style representations rather than introducing it as something entirely new.
> - **Tighter link between intuition, figures, and formalism.** The informal description of causal processes is immediately tied to Figure 2, which now explicitly shows concrete schemas such as *SwitchBurnerOn* and *FillJug* and marks which are endogenous vs. exogenous. We then refer back to these examples in the first sentence of the formal definition of causal processes.
> - **Simplified notation and explicit semantics.** In the causal-process schema we now clarify that W is a **log strength** parameter. We also define the indicator functions $C_L(\cdot)$ and $O_L(\cdot)$ before using them, and explain the role of the frame term that encourages persistence.
> - **Clear dataset notation and learning pipeline.** Section 5 now clearly distinguishes the datasets of low-level trajectories $D_{\text{low}}$ and abstract trajectories $D_{\text{abs}}$, then explains that we (i) learn parameters (Sec. 5.1), (ii) learn exogenous processes (Sec. 5.2), and (iii) learn predicates (Sec. 5.3). We also clarified the local search used in Sec. 5.3. and anchored it in prior work by adding a citation to Silver et al., 2023.
> - **Clearer explanation of variational inference and the change of basis.** In Sec. 5.1, we clarify the motivation for change of basis from delay and arrival times and describe the derivation of the ELBO at a conceptual level. The derivation and the full expression are delegated to Appendix A.3.
>
> Taken together, these changes are aimed at making the framework easier to follow without altering its technical content. We hope this alleviates the concern that the presentation is “overloaded with notation and nested definitions”. Please let us know if you see any particular concept that may benefit from better clarification.

---

> > ### Author Response · Authors · 2025-11-19
> >
> > > Insufficient experimental validation.
> > >
> >
> > Our intention was precisely to provide the kind of detailed analyses the review asks for, so we thank the reviewer for the opportunity to highlight them more clearly.
> >
> > **Breadth of evaluation.** We evaluate across **five** PyBullet tabletop domains (Coffee, Grow, Boil, Domino, Fan), each with different exogenous phenomena (filling, heating, growth, chains of collisions, airflow). For each seed, we train on 1–2 tasks with demonstrations and evaluate on **50 held-out** tasks per environment that contain more objects and more complex goals.
> >
> > **Overall performance and sample efficiency.** Figure 6(a) reports solve rate for nine agents (ours, Manual, MAPLE, ViLA zs/fs, VisPred, and four ablations), while Figure 6(b) reports online learning curves. Our method reaches near-perfect success in most domains and converges within at most three online learning iterations, while the HRL baseline (MAPLE) receives **1000× more interaction rollouts** and still struggles even on the training distribution.
> >
> > **Component-wise ablations.** To address “which components are necessary,” we include several targeted ablations:
> >
> > - **No Bayes** (no Bayesian model selection for processes),
> > - **No LLM** (replacing LLM condition proposal with a heuristic),
> > - **Manual-d** (manual abstraction but fixed, non-learned delays), and
> > - **No invent** (no predicate invention / model learning).
> >
> > Questions Q2 and Q3 in Sec. 6 explicitly analyze these variants: Manual-d and No invent show the importance of learning both model structure and delays, while No Bayes and No LLM demonstrate that Bayesian model selection and LLM guidance materially improve performance over otherwise similar pipelines.
> >
> > **Fine-grained planning statistics and qualitative analyses.** Appendix C.2 reports **per-domain planning success and wall-clock planning time** for Ours vs. Manual vs. No invent, showing that our learned model achieves similar or better success with comparable planning cost. Appendix C.1 lists the learned exogenous processes in each domain, and Figures 3 and 5 show how online learning changes the model and resulting plans across iterations.
> >
> > We agree that additional real-robot experiments would be valuable future work, but we believe the current experimental suite (5 domains, 50 held-out tasks each, 9 agents including strong RL/VLM baselines and 4 ablations, plus planning-time statistics and qualitative analyses) provides substantial evidence that the framework’s complexity is justified.

---

> ### Author Response · Authors · 2025-11-19
>
> > Conceptual confusion around endogenous vs. exogenous processes.
> >
>
> We thank the reviewer for highlighting this ambiguity. We acknowledge that "exogenous" is an overloaded term, and we have revised the text to clarify that we follow the **Temporal Planning (PDDL+)** definition rather than the Causal Inference or RL definitions.
> Confusion arises because "exogenous" implies different causal structures in different subfields:
>
> - **In Causal Inference (e.g., Pearl, 2009):** It implies a variable with no causal parents within the system (an external shock).
> - **In recent RL literature (e.g., Efroni et al., 2022):** It is strictly defined as state information that is *irrelevant* to the agent and cannot be affected by it (e.g., background video noise).
>
> In our formulation (Temporal Planning/PDDL+), **endogenous processes** represent the agent’s temporally extended skills (e.g., *SwitchBurnerOn*, *Pick*, *Place*), i.e., mechanisms directly under the agent’s control. **Exogenous processes** represent background dynamics that **unfold autonomously once certain conditions hold**, regardless of how those conditions arose (e.g., *BoilWater* once the burner is on and a pot is in place, or *FillJug* once a jug is under a running tap).
>
> Thus in the “filling water” example, **turning the faucet on** is modeled as an endogenous process, while **the subsequent rise of the water level** is an exogenous process: it continues even if the agent does nothing, and it can run concurrently with other skills. Our small-step semantics in Appendix A.1 implement this by activating exogenous processes whenever their start condition becomes true, independent of whether that condition was created by the agent or *by some external mechanism (for example*, the faucet is turned on automatically at 10 am).
>
> We have also clarified this distinction in Section 3. We hope this resolves the conceptual confusion you raised.

---

### Official Review · Reviewer_w4kc · 2025-11-03

**Soundness:** 4
**Presentation:** 4
**Contribution:** 3
**Rating:** 8
**Confidence:** 2

**Summary:**

This paper proposes to explicitly model state and temporal abstractions of *exogenous* natural processes, those not controlled by the agent directly, separately from the *endogenous* processes that are result of the agent actions. They propose an approach that leverages a VLM to process the perceptual input  and LLM to propose and filter potential predicates. They evaluate their method in simulated robotic manipulation experiments where this exogenous dynamics are part of the task.

**Strengths:**

- The paper is generally well-written and motivated
- The proposes a full approach that leverages LLMs and VLMs to improve sample efficiency in robotics planning.
- The evaluation is done in varied robotic manipulation environment and includes baselines that cover diverse approaches to solve these problems such as VLM planning, HRL and ablations to their own method.
- Their method effectively outperforms the baselines in both sample efficiency and performance
Overall this paper introduces an effective way to use modern LLMs and VLMs to create abstractions for planning that handles the processes that are not fully controlled, include natural dynamics and that have delayed effects.

**Weaknesses:**

- It is clear that searching in the space of predicates is extremely costly and intractable via brute-force. It seems that LLMs provide a way to amortize the cost of it. However, it still seems a very large search problem. How computationally expensive is the method in comparison to some of the baselines?
- The notation is quite cluttered and it's a bit hard to parse, though it can be understood. Perhaps some rework on simplifying notation could help the reader and I'd encourage the authors to do it.

**Questions:**

See above

---

> ### Author Response · Authors · 2025-11-19
>
> We thank the reviewer for the encouraging assessment and helpful suggestions.
>
> > It is clear that searching in the space of predicates is extremely costly and intractable via brute-force. It seems that LLMs provide a way to amortize the cost of it. However, it still seems a very large search problem. How computationally expensive is the method in comparison to some of the baselines?
> >
>
> A full brute-force search over predicates would indeed be intractable. In practice, we avoid this by combining LLM proposals with a *local hill-climbing* search in predicate space (Silver et al., 2023): starting from the current predicate set, we accept only those that improve the learning objective, stopping once no further improvement is observed. This keeps the explored neighborhood small and far from an exhaustive search.
>
> We have now also made the computational cost explicit in Appendix C.2. Across domains, a single online learning iteration of **ExoPredicator** takes roughly 30–360 seconds of wall-clock time on 32 CPU cores (no GPU). By comparison, the **HRL (MAPLE)** baseline requires on average about 600 seconds per online iteration, and its runtime grows substantially as it gathers more data, reaching up to ~3200 seconds in some runs. The **VLM planning (ViLA)** baseline does not perform online learning, but can incur substantial *test-time* cost because it issues expensive VLM calls every time it generates a plan. Appendix C.2 also reports per-episode planning times: once trained, ExoPredicator’s planning time is of the same order as the Manual oracle (e.g., around 1 second in Coffee/Grow and tens of seconds in Domino/Fan). Because the agent converges in at most three online iterations in all domains, the overall computational cost remains moderate.
>
> > The notation is quite cluttered and it's a bit hard to parse, though it can be understood. Perhaps some rework on simplifying notation could help the reader and I'd encourage the authors to do it.
> >
>
> We agree and have revised the exposition to make the notation easier to follow:
>
> - **Reorganized exposition around a running example.** Section 3 now first explains the causal-process framework in prose using the running example in Figure 2, and only then introduces the formal schema and probabilistic semantics. This makes it easier to map each symbol to a concrete example.
> - **Simplified notation and explicit semantics.** In the causal-process schema we now state explicitly that $W$ is a *log strength* parameter. We define the indicator functions $C_L(\cdot)$ and $O_L(\cdot)$ before using them, and describe the role of the frame potential that encourages state persistence over time.
> - **Clear dataset notation and learning pipeline.** Section 5 now clearly distinguishes low-level trajectories $D_{\text{low}}$ from abstract trajectories $D_{\text{abs}}$, and explicitly describes the learning pipeline: (i) parameter learning (Sec. 5.1), (ii) exogenous process learning (Sec. 5.2), and (iii) predicate learning (Sec. 5.3). We also clarify the local search used in Sec. 5.3 and anchor it in prior work with a citation to Silver et al. (2023).
> - **Clearer explanation of variational inference and change of basis.** In Sec. 5.1 we now explain, at a conceptual level, the change of basis from delays to arrival times and the resulting ELBO decomposition, while moving the full derivation and final expression to Appendix A.3 to reduce clutter in the main text.
> - **Clarified Variational Distribution:** We have explicitly clarified the form and parameterization of both the delay distribution and the variational distribution in **Section 5.1**, **Appendix A.3**, and **Appendix C.3** to ensure the probabilistic model is fully defined.
>
> We believe these changes make the framework much more accessible and thank you for prompting this improvement.

---

### Meta-Review · Area_Chair_xUSd · 2026-01-07

**Summary:**

The paper received strong endorsement from three of the four reviewers, who recognized its timely contribution to long-horizon robotic planning in dynamic environments with exogenous processes. In contrast, one reviewer found the presentation overly complex and questioned both the clarity of the formalism and the justification for the framework’s intricacy. The author provided a detailed response and made modifications and explanations for each weakness and question. In the end, the author made extensive revisions in the latest manuscript of the paper, fully addressing every question and weakness raised.

**Reviewer Concerns:**

The authors addressed all concerns raised by every reviewer in their rebuttal. They significantly revised the manuscript to improve quality.

**Reviewer Scores:**

Reviewer w4kc (initial 8) would likely maintain their score, as their minor concerns about computational cost and notation were fully addressed.
Reviewer Epsw (initial 8) would likely maintain their score, given the thorough responses to questions on delay modeling.
Reviewer p9tB (initial 2) might raise their score modestly due to the substantial revisions improving clarity and experimental detail.
Reviewer Fikj (initial 8) would likely maintain their score, as their constructive questions on variational inference, LLM prompting, and MDL priors were clearly answered.

---

### Decision · Program_Chairs · 2026-01-26

Accept (Poster)